# BEHAVIOR PROXIMAL POLICY OPTIMIZATION

**Zifeng Zhuang**[12*]  **Kun Lei**[2*]  **Jinxin Liu**[2]  **Donglin Wang**[23†]  **Yilang Guo**[4]

[1] Zhejiang University.   [2] School of Engineering, Westlake University.

[3] Institute of Advanced Technology, Westlake Institute for Advanced Study.

[4] School of Software Engineering, Beijing Jiaotong University.

`{zhuangzifeng,leikun,liujinxin,wangdonglin}@westlake.edu.cn,`
`20301130@bjtu.edu.cn`

## ABSTRACT

Offline reinforcement learning (RL) is a challenging setting where existing off-policy actor-critic methods perform poorly due to overestimating of out-of-distribution state-action pairs. Thus, various additional augmentations are proposed to keep the learned policy close to the offline dataset (or the behavior policy). In this work, starting from the analysis of offline monotonic policy improvement, we reach a surprising conclusion that *online on-policy algorithms are naturally able to solve offline RL*. Specifically, the inherent conservatism of these on-policy algorithms is exactly what the offline RL method needs to overcome the overestimation. Based on this, we propose **B**ehavior **P**roximal **P**olicy **O**ptimization (BPPO), which solves offline RL without any extra constraint or regularization introduced compared to PPO. Extensive experiments on the D4RL benchmark empirically show this extremely succinct method outperforms state-of-the-art offline RL algorithms. Our implementation is available at `https://github.com/Dragon-Zhuang/BPPO`.

## 1 INTRODUCTION

Typically, reinforcement learning (RL) is thought of as a paradigm for online learning, where the agent interacts with the environment to collect experiences and then uses them to improve itself (Sutton et al., 1998). This online process poses the biggest obstacles to real-world RL applications because of expensive or even risky data collection in some fields (such as navigation (Mirowski et al., 2018) and healthcare (Yu et al., 2021a)). As an alternative, offline RL eliminates the online interaction and learns from a fixed dataset collected by some arbitrary and possibly unknown process (Lange et al., 2012; Fu et al., 2020). The prospect of this data-driven mode (Levine et al., 2020) is pretty encouraging and has been placed with great expectations for solving real-world RL applications.

Unfortunately, the major superiority of offline RL, the lack of online interaction, also raises another challenge. The classical off-policy iterative algorithms tend to underperform due to overestimating out-of-distribution (shorted as OOD) state-action pairs, even though offline RL can be viewed as an extreme off-policy case. More specifically, when $Q$-function poorly estimates the value of OOD state-action pairs during policy evaluation, the agent tends to take OOD actions with erroneously estimated high values, resulting in low-performance after policy improvement (Fujimoto et al., 2019). Thus, to overcome the overestimation issue, some solutions keep the learned policy close to the behavior policy (or the offline dataset) (Fujimoto et al., 2019; Wu et al., 2019; Fujimoto & Gu, 2021).

Most offline RL algorithms adopt online interactions to select hyperparameters. This is because offline hyperparameter selection, which selects hyperparameters without online interactions, is always an open problem lacking satisfactory solutions (Paine et al., 2020; Zhang & Jiang, 2021). Deploying the policy learned by offline RL is potentially risky in certain areas (Mirowski et al., 2018; Yu et al., 2021a) since the performance is unknown. However, the risk during online interactions will be greatly reduced if the deployed policy can guarantee better performance than the behavior policy. This inspires us to consider how to use offline dataset to improve behavior policy with a monotonic performance guarantee. We formulate this problem as offline monotonic policy improvement.

---

[*] Equal contribution.

[†] Corresponding author.

To analyze offline monotonic policy improvement, we introduce the Performance Difference Theorem (Kakade & Langford, 2002). During analysis, we find that the offline setting does make the monotonic policy improvement more complicated, but the way to monotonically improve policy remains unchanged. This indicates the algorithms derived from *online* monotonic policy improvement (such as Proximal Policy Optimization) can also achieve *offline* monotonic policy improvement. In other words, PPO can naturally solve offline RL. Based on this surprising discovery, we propose **B**ehavior **P**roximal **P**olicy **O**ptimization (BPPO), an offline algorithm that monotonically improves behavior policy in the manner of PPO. Owing to the inherent conservatism of PPO, BPPO restricts the ratio of learned policy and behavior policy within a certain range, similar to the offline RL methods which make the learned policy close to the behavior policy. As offline algorithms are becoming more and more sophisticated, TD3+BC (Fujimoto & Gu, 2021), which augments TD3 (Fujimoto et al., 2018) with behavior cloning (Pomerleau, 1988), reminds us to revisit the simple alternatives with potentially good performance. BPPO is such a "most simple" alternative without introducing any extra constraint or regularization on the basis of PPO. Extensive experiments on the D4RL benchmark (Fu et al., 2020) empirically shows that BPPO outperforms state-of-the-art offline RL algorithms.

## 2 PRELIMINARIES

### 2.1 REINFORCEMENT LEARNING

Reinforcement Learning (RL) is a framework of sequential decision. Typically, this problem is formulated by a Markov decision process (MDP) $\mathcal{M} = \{\mathcal{S}, \mathcal{A}, r, p, d_0, \gamma\}$, with state space $\mathcal{S}$, action space $\mathcal{A}$, scalar reward function $r$, transition dynamics $p$, initial state distribution $d_0(s_0)$ and discount factor $\gamma$ (Sutton et al., 1998). The objective of RL is to learn a policy, which defines a distribution over action conditioned on states $\pi(a_t|s_t)$ at timestep $t$, where $a_t \in \mathcal{A}, s_t \in \mathcal{S}$. Given this definition, the trajectory $\tau = (s_0, a_0, \cdots, s_T, a_T)$ generated by the agent's interaction with environment $\mathcal{M}$ can be described as a distribution $P_\pi(\tau) = d_0(s_0) \prod_{t=0}^{T} \pi(a_t|s_t) p(s_{t+1}|s_t, a_t)$, where $T$ is the length of the trajectory, and it can be infinite. Then, the goal of RL can be written as an expectation under the trajectory distribution $J(\pi) = \mathbb{E}_{\tau \sim P_\pi(\tau)} \left[ \sum_{t=0}^{T} \gamma^t r(s_t, a_t) \right]$. This objective can also be measured by a state-action value function $Q_\pi(s, a)$, the expected discounted return given the action $a$ in state $s$: $Q_\pi(s, a) = \mathbb{E}_{\tau \sim P_\pi(\tau|s,a)} \left[ \sum_{t=0}^{T} \gamma^t r(s_t, a_t)|s_0 = s, a_0 = a \right]$. Similarly, the value function $V_\pi(s)$ is the expected discounted return of a certain state $s$: $V_\pi(s) = \mathbb{E}_{\tau \sim P_\pi(\tau|s)} \left[ \sum_{t=0}^{T} \gamma^t r(s_t, a_t)|s_0 = s \right]$. Then, we can define the advantage function: $A_\pi(s, a) = Q_\pi(s, a) - V_\pi(s)$.

### 2.2 OFFLINE REINFORCEMENT LEARNING

In offline RL, the agent only has access to a fixed dataset with transitions $\mathcal{D} = \left\{ (s_t, a_t, s_{t+1}, r_t)_{t=1}^{N} \right\}$ collected by a behavior policy $\pi_\beta$. Without interacting with environment $\mathcal{M}$, offline RL expects the agent to infer a policy from the dataset. Behavior cloning (BC) (Pomerleau, 1988), an approach of imitation learning, can directly imitate the action of each state with supervised learning:

$$\hat{\pi}_\beta = \underset{\pi}{\operatorname{argmax}} \, \mathbb{E}_{(s,a) \sim \mathcal{D}} \left[ \log \pi(a|s) \right]. \tag{1}$$

Note that the performance of $\hat{\pi}_\beta$ trained by behavior cloning highly depends on the quality of transitions, also the collection process of behavior policy $\pi_\beta$. In the rest of this paper, improving behavior policy actually refers to improving the estimated behavior policy $\hat{\pi}_\beta$, because $\pi_\beta$ is unknown.

### 2.3 PERFORMANCE DIFFERENCE THEOREM

**Theorem 1.** *(Kakade & Langford, 2002) Let the discounted unnormalized visitation frequencies as* $\rho_\pi(s) = \sum_{t=0}^{T} \gamma^t P(s_t = s|\pi)$ *and* $P(s_t = s|\pi)$ *represents the probability of the $t$-th state equals to $s$ in trajectories generated by policy $\pi$. For any two policies $\pi$ and $\pi'$, the performance difference* $J_\Delta(\pi', \pi) \triangleq J(\pi') - J(\pi)$ *can be measured by the advantage function:*

$$J_\Delta(\pi', \pi) = \mathbb{E}_{\tau \sim P_{\pi'}(\tau)} \left[ \sum_{t=0}^{T} \gamma^t A_\pi(s_t, a_t) \right] = \mathbb{E}_{s \sim \rho_{\pi'}(\cdot), a \sim \pi'(\cdot|s)} \left[ A_\pi(s, a) \right]. \tag{2}$$

Derivation detail is presented in Appendix A. This theorem implies that improving policy from $\pi$ to $\pi'$ can be achieved by maximizing (2). From this theorem, Trust Region Policy Optimization (TRPO) (Schulman et al., 2015a) is derived, which can guarantee the monotonic improvement of performance. We also apply this theorem to formulate offline monotonic policy improvement.

## 3  OFFLINE MONOTONIC IMPROVEMENT OVER BEHAVIOR POLICY

In this section, we theoretically analyze offline monotonic policy improvement based on Theorem 1, namely improving the $\hat{\pi}_\beta$ generated by behavior cloning (1) with offline dataset $\mathcal{D}$. Applying the Performance Difference Theorem to the estimated behavior policy $\hat{\pi}_\beta$, we can get

$$J_\Delta\left(\pi, \hat{\pi}_\beta\right) = \mathbb{E}_{s \sim \rho_\pi(\cdot), a \sim \pi(\cdot|s)}\left[A_{\hat{\pi}_\beta}(s, a)\right]. \tag{3}$$

Maximizing this equation can obtain a policy better than behavior policy $\hat{\pi}_\beta$. But the above equation is not tractable due to the dependence of the new policy's state distribution $\rho_\pi(s)$. For standard online method, $\rho_\pi(s)$ is replaced by the old state distribution $\rho_{\hat{\pi}_\beta}(s)$. But in the offline setting, $\rho_{\hat{\pi}_\beta}(s)$ cannot be obtained through interactions with the environment like the online situation. We use the state distribution recovered by the offline dataset $\rho_\mathcal{D}(s)$ for replacement, where $\rho_\mathcal{D}(s) = \sum_{t=0}^{T} \gamma^t P\left(s_t = s|\mathcal{D}\right)$ and $P\left(s_t = s|\mathcal{D}\right)$ represents the probability of the $t$-th state equals to $s$ in the offline dataset. Therefore, the approximation of $J_\Delta\left(\pi, \pi_\beta\right)$ can be written as:

$$\widehat{J}_\Delta\left(\pi, \hat{\pi}_\beta\right) = \mathbb{E}_{s \sim \rho_\mathcal{D}(\cdot), a \sim \pi(\cdot|s)}\left[A_{\hat{\pi}_\beta}(s, a)\right]. \tag{4}$$

To measure the difference between $J_\Delta\left(\pi, \hat{\pi}_\beta\right)$ and its approximation $\widehat{J}_\Delta\left(\pi, \hat{\pi}_\beta\right)$, we introduce a midterm $\mathbb{E}_{s \sim \rho_{\hat{\pi}_\beta}(s), a \sim \pi(\cdot|s)}\left[A_{\hat{\pi}_\beta}(s, a)\right]$ with the state distribution $\rho_{\hat{\pi}_\beta}(s)$. During the proof, the commonly-used total variational divergence $D_{TV}\left(\pi\|\hat{\pi}_\beta\right)[s] = \frac{1}{2}\mathbb{E}_a|\pi(a|s) - \hat{\pi}_\beta(a|s)|$ between policy $\pi, \hat{\pi}_\beta$ at state $s$ is necessary. For the total variational divergence between the offline dataset $\mathcal{D}$ and the estimated behavior policy $\hat{\pi}_\beta$, it may not be straightforward. We can view the offline dataset $\mathcal{D} = \left\{(s_t, a_t, s_{t+1}, r_t)_{t=1}^{N}\right\}$ as a deterministic distribution, and then the distance is:

**Proposition 1.** *For offline dataset $\mathcal{D} = \left\{(s_t, a_t, s_{t+1}, r_t)_{t=1}^{N}\right\}$ and policy $\hat{\pi}_\beta$, the total variational divergence can be expressed as $D_{TV}\left(\mathcal{D}\|\hat{\pi}_\beta\right)[s_t] = \frac{1}{2}\left(1 - \hat{\pi}_\beta(a_t|s_t)\right)$.*

Detailed derivation process is presented in Appendix B. Now we are ready to measure the difference:

**Theorem 2.** *Given the distance $D_{TV}\left(\pi\|\hat{\pi}_\beta\right)[s]$ and $D_{TV}\left(\mathcal{D}\|\hat{\pi}_\beta\right)[s] = \frac{1}{2}\left(1 - \hat{\pi}_\beta(a_t|s_t)\right)$, we can derive the following bound:*

$$J_\Delta\left(\pi, \hat{\pi}_\beta\right) \geq \widehat{J}_\Delta\left(\pi, \hat{\pi}_\beta\right) - 4\gamma\mathbb{A}_{\hat{\pi}_\beta} \cdot \max_s D_{TV}\left(\pi\|\hat{\pi}_\beta\right)[s] \cdot \mathbb{E}_{s \sim \rho_{\hat{\pi}_\beta}(\cdot)}\left[D_{TV}\left(\pi\|\hat{\pi}_\beta\right)[s]\right]$$

$$- 2\gamma\mathbb{A}_{\hat{\pi}_\beta} \cdot \max_s D_{TV}\left(\pi\|\hat{\pi}_\beta\right)[s] \cdot \mathbb{E}_{s \sim \rho_\mathcal{D}(\cdot)}\left[1 - \hat{\pi}_\beta(a|s)\right], \tag{5}$$

*here $\mathbb{A}_{\hat{\pi}_\beta} = \max_{s,a}\left|A_{\hat{\pi}_\beta}(s, a)\right|$. The proof is presented in Appendix C.*

Compared to the theorem in the online setting (Schulman et al., 2015a; Achiam et al., 2017; Queeney et al., 2021), the second right term of Equation (5) is similar while the third term is unique for the offline. $\mathbb{E}_{s \sim \rho_\mathcal{D}(\cdot)}\left[1 - \hat{\pi}_\beta(a|s)\right]$ represents the difference caused by the mismatch between offline dataset $\mathcal{D}$ and $\hat{\pi}_\beta$. When $\hat{\pi}_\beta$ is determined, this term is one constant. And because the inequality $\max_s D_{TV}\left(\pi\|\hat{\pi}_\beta\right)[s] \geq \mathbb{E}_{s \sim \rho_{\hat{\pi}_\beta}(\cdot)}\left[D_{TV}\left(\pi\|\hat{\pi}_\beta\right)[s]\right]$ holds, we can claim the following conclusion:

> **Conclusion 1**
>
> To guarantee the true objective $J_\Delta\left(\pi, \hat{\pi}_\beta\right)$ non-decreasing, we should simultaneously maximize $\mathbb{E}_{s \sim \rho_\mathcal{D}(\cdot), a \sim \pi(\cdot|s)}\left[A_{\hat{\pi}_\beta}(s, a)\right]$ and minimize $\left[\max_s D_{TV}\left(\pi\|\hat{\pi}_\beta\right)[s]\right]$, which means the offline dataset $\mathcal{D}$ is capable of monotonically improving the estimated behavior policy $\hat{\pi}_\beta$.

Suppose we have improved the behavior policy $\hat{\pi}_\beta$ and get a policy $\pi_k$. The above theorem only guarantees that $\pi_k$ has a higher performance than $\hat{\pi}_\beta$ but $\pi_k$ may not be optimal. If the offline dataset $\mathcal{D}$ can still improve the policy $\pi_k$ to get a better policy $\pi_{k+1}$, $\pi_{k+1}$ must be closer to the optimal policy. Thus, we further analyze the monotonic policy improvement over policy $\pi_k$. Applying Performance Difference Theorem 1 to the policy $\pi_k$,

$$J_\Delta\left(\pi, \pi_k\right) = \mathbb{E}_{s \sim \rho_\pi(\cdot), a \sim \pi(\cdot|s)}\left[A_{\pi_k}(s, a)\right]. \tag{6}$$

To approximate the above equation, the common manner is replacing $\rho_\pi$ with the old policy state distribution $\rho_{\pi_k}$. But in the offline RL, $\pi_k$ is forbidden from acting in the environment. As a result, the state distribution $\rho_{\pi_k}$ is impossible to estimate. Thus, the only choice without any other alternative is replacing $\rho_{\pi_k}$ by the state distribution from the offline dataset $\mathcal{D}$:

$$\widehat{J}_\Delta\left(\pi, \pi_k\right) = \mathbb{E}_{s \sim \rho_\mathcal{D}(\cdot), a \sim \pi(\cdot|s)}\left[A_{\pi_k}(s, a)\right]. \tag{7}$$

Intuitively, this replacement is reasonable if $\pi_k, \hat{\pi}_\beta$ are similar which means this approximation must be related to the distance $D_{TV}\left(\pi_k \| \hat{\pi}_\beta\right)[s]$. Concretely, the gap can be formulated as follows:

**Theorem 3.** *Given the distance $D_{TV}\left(\pi \| \pi_k\right)[s]$, $D_{TV}\left(\pi_k \| \hat{\pi}_\beta\right)[s]$ and $D_{TV}\left(\mathcal{D} \| \hat{\pi}_\beta\right)[s] = \frac{1}{2}\left(1 - \hat{\pi}_\beta(a|s)\right)$, we can derive the following bound:*

$$\begin{aligned}
J_\Delta\left(\pi, \pi_k\right) \geq \widehat{J}_\Delta\left(\pi, \pi_k\right) &- 4\gamma \mathbb{A}_{\pi_k} \cdot \max_s D_{TV}\left(\pi \| \pi_k\right)[s] \cdot \mathbb{E}_{s \sim \rho_{\pi_k}(\cdot)}\left[D_{TV}\left(\pi \| \pi_k\right)[s]\right] \\
&- 4\gamma \mathbb{A}_{\pi_k} \cdot \max_s D_{TV}\left(\pi \| \pi_k\right)[s] \cdot \mathbb{E}_{s \sim \rho_{\hat{\pi}_\beta}(\cdot)}\left[D_{TV}\left(\pi_k \| \hat{\pi}_\beta\right)[s]\right] \\
&- 2\gamma \mathbb{A}_{\pi_k} \cdot \max_s D_{TV}\left(\pi \| \pi_k\right)[s] \cdot \mathbb{E}_{s \sim \rho_\mathcal{D}(\cdot)}\left[1 - \hat{\pi}_\beta(a|s)\right],
\end{aligned} \tag{8}$$

*here $\mathbb{A}_{\pi_k} = \max_{s,a}|A_{\pi_k}(s, a)|$. The proof is presented in Appendix D.*

Compared to the theorem 2, one additional term related to the distance of $\pi_k, \hat{\pi}_\beta$ has been introduced. The distance $\mathbb{E}_{s \sim \rho_{\hat{\pi}_\beta}(\cdot)}\left[D_{TV}\left(\pi_k \| \hat{\pi}_\beta\right)[s]\right]$ is irrelevant to the target policy $\pi$ which can also be viewed as one constant. Besides, theorem 2 is a specific case of this theorem if $\pi_k = \hat{\pi}_\beta$. Thus, we set $\boxed{\pi_0 = \hat{\pi}_\beta}$ since $\hat{\pi}_\beta$ is the first policy to be improved and in the following section we will no longer deliberately distinguish $\hat{\pi}_\beta, \pi_k$. Similarly, we can derive the following conclusion:

---

**Conclusion 2**

To guarantee the true objective $J_\Delta\left(\pi, \pi_k\right)$ non-decreasing, we should simultaneously maximize $\mathbb{E}_{s \sim \rho_\mathcal{D}(\cdot), a \sim \pi(\cdot|s)}\left[A_{\pi_k}(s, a)\right]$ and minimize $\left[\max_s D_{TV}\left(\pi \| \pi_k\right)[s]\right]$, which means the offline dataset $\mathcal{D}$ is capable of monotonically improving the policy $\pi_k$, where $k = 0, 1, 2, \cdots$.

---

## 4  BEHAVIOR PROXIMAL POLICY OPTIMIZATION

In this section, we derive a practical algorithm based on the theoretical results. And surprisingly, the loss function of this algorithm is the same as the online on-policy method Proximal Policy Optimization (PPO) (Schulman et al., 2017). Furthermore, this algorithm highly depends on the behavior policy so we name it as **B**ehavior **P**roximal **P**olicy **O**ptimization, shorted as BPPO.

According to the **Conclusion 2**, to monotonically improve policy $\pi_k$, we should jointly optimize:

$$\mathbf{Maximize}_\pi \ \mathbb{E}_{s \sim \rho_\mathcal{D}(\cdot), a \sim \pi(\cdot|s)}\left[A_{\pi_k}(s, a)\right] \ \& \ \mathbf{Minimize}_\pi \ \max_s D_{TV}\left(\pi \| \pi_k\right)[s], \tag{9}$$

here $k = 0, 1, 2, \cdots$ and $\pi_0 = \hat{\pi}_\beta$. But minimizing the total divergence between $\pi$ and $\pi_k$ results in a trivial solution $\pi = \pi_k$ which is impossible to make improvement over $\pi_k$. A more reasonable optimization objective is to maximize $\widehat{J}_\Delta\left(\pi, \pi_k\right)$ while constraining the divergence:

$$\mathbf{Maximize}_\pi \ \mathbb{E}_{s \sim \rho_\mathcal{D}(\cdot), a \sim \pi(\cdot|s)}\left[A_{\pi_k}(s, a)\right] \ \text{s.t.} \ \max_s D_{TV}\left(\pi \| \pi_k\right)[s] \leq \epsilon. \tag{10}$$

For the term to be maximized, we adopt importance sampling to make the expectation only depends on the action distribution of the old policy $\pi_k$ rather than the new policy $\pi$:

$$\mathbb{E}_{s\sim\rho_\mathcal{D}(\cdot),a\sim\pi(\cdot|s)}\left[A_{\pi_k}(s,a)\right] = \mathbb{E}_{s\sim\rho_\mathcal{D}(\cdot),a\sim\pi_k(\cdot|s)}\left[\frac{\pi(a|s)}{\pi_k(a|s)}A_{\pi_k}(s,a)\right]. \tag{11}$$

In this way, we could estimate this term by sampling states from offline the dataset $s\sim\rho_\mathcal{D}(\cdot)$ then sampling actions with old policy $a\sim\pi_k(\cdot|s)$. For the total variational divergence, we rewrite it as

$$\max_s D_{TV}(\pi\|\pi_k)[s] = \max_s \frac{1}{2}\int_a |\pi(a|s) - \pi_k(a|s)|\,\mathrm{d}a$$

$$= \max_s \frac{1}{2}\int_a \pi_k(a|s)\left|\frac{\pi(a|s)}{\pi_k(a|s)} - 1\right|\mathrm{d}a = \frac{1}{2}\max_s \mathbb{E}_{a\sim\pi_k(\cdot|s)}\left|\frac{\pi(a|s)}{\pi_k(a|s)} - 1\right|. \tag{12}$$

In the offline setting, only states $s\sim\rho_\mathcal{D}(\cdot)$ are available and other states are inaccessible. So the operation $\max_s$ can also be expressed as $\max\limits_{s\sim\rho_\mathcal{D}(\cdot)}$. When comparing Equation (11) and (12), we find that the state distribution, the action distribution and the policy ratio appear in both. Thus we consider how to insert the divergence constraint into Equation (11). The following constraints are equivalent:

$$\max_{s\sim\rho_\mathcal{D}(\cdot)} D_{TV}(\pi\|\pi_k)[s] \le \epsilon \iff \max_{s\sim\rho_\mathcal{D}(\cdot)} \mathbb{E}_{a\sim\pi_k(\cdot|s)}\left|\frac{\pi(a|s)}{\pi_k(a|s)} - 1\right| \le 2\epsilon$$

$$\iff \max_{s\sim\rho_\mathcal{D}(\cdot)} \mathbb{E}_{a\sim\pi_k(\cdot|s)}\,\mathrm{clip}\left(\frac{\pi(a|s)}{\pi_k(a|s)}, 1 - 2\epsilon, 1 + 2\epsilon\right), \mathrm{clip}(x,l,u) = \min(\max(x,l),u). \tag{13}$$

Here the $\max$ operation is impractical to solve, so we adopt a heuristic approximation (Schulman et al., 2015a) that changes $\max$ into expectation. Then divergence constraint (13) can be inserted:

$$L_k(\pi) = \mathbb{E}_{s\sim\rho_\mathcal{D}(\cdot),a\sim\pi_k(\cdot|s)}\left[\min\left(\frac{\pi(a|s)}{\pi_k(a|s)}A_{\pi_k}(s,a), \mathrm{clip}\left(\frac{\pi(a|s)}{\pi_k(a|s)}, 1 - 2\epsilon, 1 + 2\epsilon\right)A_{\pi_k}(s,a)\right)\right], \tag{14}$$

where the operation $\min$ makes this objective become the lower bound of Equation (11). This loss function is quite similar to PPO (Schulman et al., 2017) and the only difference is the state distribution. Therefore, we claim that *online on-policy algorithms are naturally able to solve offline RL.*

## 5 DISCUSSIONS AND IMPLEMENTATION DETAILS

In this section, we first directly highlight why BPPO can solve offline reinforcement learning, namely, how to overcome the overestimation issue. Then we discuss some implementation details, especially, the approximation of the advantage $A_{\pi_k}(s,a)$. Finally, we analyze the relation between BPPO and previous algorithms including Onestep RL and iterative methods.

**Why BPPO can solve offline RL?** According to the final loss (14) and Equation (13), BPPO actually constrains the closeness by the expectation of the total variational divergence:

$$\mathbb{E}_{s\sim\rho_\mathcal{D}(\cdot),a\sim\pi_k(\cdot|s)}\left|\frac{\pi(a|s)}{\pi_k(a|s)} - 1\right| \le 2\epsilon. \tag{15}$$

If $k = 0$, this equation ensures the closeness between learned policy $\pi$ and behavior policy $\hat{\pi}_\beta$. When $k > 0$, one issue worthy of attention is **whether the closeness between learned policy $\pi$ and $\pi_k$ can indirectly constrain the closeness between $\pi$ and $\hat{\pi}_\beta$**. To achieve this, also to prevent the learned policy $\pi$ completely away from $\hat{\pi}_\beta$, we introduce a technique called clip ratio decay. As the policy updates, the clip ratio $\epsilon$ gradually decreases until reaching a certain training step (such as 200 steps):

$$\epsilon_i = \epsilon_0 \times (\sigma)^i \;\textbf{IF}\; i \le 200 \;\textbf{ELSE}\; \epsilon_i = \epsilon_{200} \tag{16}$$

here $i$ denotes the training steps, $\epsilon_0$ denotes the initial clip ratio, and $\sigma \in (0, 1]$ is the decay coefficient.

From Figure 1(a) and 1(b), we can find that the ratio $\pi_k/\hat{\pi}_\beta$ may be out of the certain range $[1 - 2\epsilon, 1 + 2\epsilon]$ (the region surrounded by the dotted pink and purple line) without clip ratio decay technique (also $\sigma = 1$). But the ratio stays within the range stably when the decay is applied which means the Equation (15) can ensure the closeness between the final learned policy by BPPO and behavior policy.

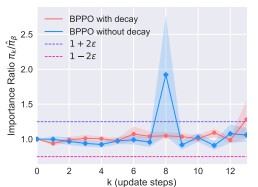
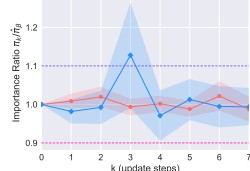

(a) hopper-medium      (b) hopper-medium-replay

Figure 1: Visualization of the importance weight between the updated policy $\pi_k$ and the estimated behavior policy $\hat{\pi}_\beta$.

**How to approximate the advantage?** When calculating the loss function (14), the only difference from the online situation is the approximation of advantage $A_{\pi_k}(s, a)$. In online RL, GAE (Generalized Advantage Estimation) (Schulman et al., 2015b) approximates the $A_{\pi_k}$ using the data collected by policy $\pi_k$. Obviously, GAE is inappropriate in the offline situations due to the existence of online interaction. As a result, BPPO has to calculate the advantage $A_{\pi_k} = Q_{\pi_k} - V_{\pi_\beta}$ in an off-policy manner where $Q_{\pi_k}$ is calculated by Q-learning (Watkins & Dayan, 1992) using offline dataset $\mathcal{D}$ and $V_{\pi_\beta}$ is calculated by fitting returns $\sum_{t=0}^{T} \gamma^t r(s_t, a_t)$ using the MSE loss. Note that the value function is $V_{\pi_\beta}$ rather than $V_{\pi_k}$ since the state distribution has been changed into $s \sim \rho_\mathcal{D}(\cdot)$ in Theorem 2, 3.

---

**Algorithm 1** **B**ehavior **P**roximal **P**olicy **O**ptimization (BPPO)

1: Estimate behavior policy $\hat{\pi}_\beta$ by behavior cloning;
2: Calculate $Q$-function $Q_{\pi_\beta}$ by SARSA;
3: Calculate value function $V_{\pi_\beta}$ by fitting returns;
4: Initialize $k = 0$ and set $\pi_k \leftarrow \pi_\beta$ & $Q_{\pi_k} = Q_{\pi_\beta}$;
5: **for** $i = 0, 1, 2, \cdots, I$ **do**
6:     $A_{\pi_k} = Q_{\pi_k} - V_{\pi_\beta}$
7:     Update the policy $\pi$ by maximizing $L_k(\pi)$;
8:     **if** $J(\pi) > J(\pi_k)$ **then**
9:         Set $k = k + 1$ & $\pi_k \leftarrow \pi$;
10:         **if** *advantage replacement* **then**
11:             $Q_{\pi_k} = Q_{\pi_\beta}$;
12:         **else**
13:             Calculate $Q_{\pi_k}$ by Q-learning;
14:         **end if**
15:     **end if**
16: **end for**

---

Besides, we have another simple choice based on the results that $\pi_k$ is close to the $\pi_\beta$ with the help of clip ratio decay. We can replace all the $A_{\pi_k}$ with the $A_{\pi_\beta}$, which may introduce some error but the benefit is that $A_{\pi_\beta}$ must be more accurate than $A_{\pi_k}$ since off-policy estimation is potentially dangerous, especially in the offline setting. We conduct a series of experiments in Section 7.2 to compare these two implementations and find that the latter one, *advantage replacement*, is better. Based on the above implementation details, we summarize the whole workflow of BPPO in Algorithm 1.

**What is the relation between BPPO, Onestep RL and iterative methods?** Since BPPO is highly related to on-policy algorithms, it is naturally associated with Onestep RL (Brandfonbrener et al., 2021) that solves offline RL without off-policy evaluation. If we remove lines 8~15 in Algorithm 1, we get Onestep version of BPPO, which means only the behavior policy $\hat{\pi}_\beta$ is improved. In contrast, BPPO also improves $\pi_k$, the policy that has been improved over $\hat{\pi}_\beta$. The right figure shows the difference between BPPO and its Onestep version: Onestep strictly requires the new policy close to $\hat{\pi}_\beta$, while BPPO appropriately loosens this restriction.

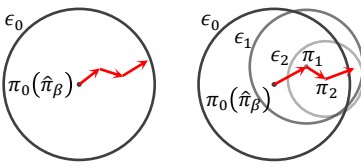

Figure 2: The difference between Onestep BPPO (left) and BPPO (right), where the decreasing circle corresponds to $\epsilon$ decay.

If we calculate the $Q$-function in off-policy manner, namely, line 13 in Algorithm 1, the method switches to an iterative style. If we adopt *advantage replacement*, line 11, BPPO only estimates the advantage function once but updates many policies, from $\hat{\pi}_\beta$ to $\pi_k$. Onestep RL estimates the $Q$-function once and use it to update estimated behavior policy. Iterative methods estimate $Q$-function several times and then update the corresponding policy. Strictly speaking, BPPO is neither an Onestep nor an iterative method. BPPO is a special case between these two types.

## 6 RELATED WORK

**Offline Reinforcement Learning**   Most of the online off-policy methods fail or underperform in offline RL due to extrapolation error (Fujimoto et al., 2019) or distributional shift (Levine et al., 2020). Thus most offline algorithms typically augment existing off-policy algorithms with a penalty measuring divergence between the policy and the offline data (or behavior policy). Depending on how to implement this penalty, a variety of methods were proposed such as batch constrained (Fujimoto et al., 2019), KL-control (Jaques et al., 2019; Liu et al., 2022b), behavior-regularized (Wu et al., 2019; Fujimoto & Gu, 2021) and policy constraint (Kumar et al., 2019; Levine et al., 2020; Kostrikov et al., 2021). Other methods augment BC with a weight to make the policy favor high advantage actions (Wang et al., 2018; Siegel et al., 2020; Peng et al., 2019; Wang et al., 2020). Some methods extra introduced Uncertainty estimation (An et al., 2021b; Bai et al., 2022) or conservative (Kumar et al., 2020; Yu et al., 2021b; Nachum et al., 2019) estimation to overcome overestimation.

**Monotonic Policy Improvement**   Monotonic policy improvement in online RL was first introduced by Kakade & Langford (2002). On this basis, two classical on-policy methods Trust Region Policy Optimization (TRPO) (Schulman et al., 2015a) and Proximal Policy Optimization (PPO) (Schulman et al., 2017) were proposed. Afterwards, monotonic policy improvement has been extended to constrained MDP (Achiam et al., 2017), model-based method (Luo et al., 2018) and off-policy RL (Queeney et al., 2021; Meng et al., 2021). The main idea behind BPPO is to regularize each policy update by restricting the divergence. Such regularization is often used in unsupervised skill learning (Liu et al., 2021; 2022a; Tian et al., 2021) and imitation learning (Xiao et al., 2019; Kang et al., 2021). Xu et al. (2021) mentions that offline algorithms lack guaranteed performance improvement over the behavior policy but we are the first to introduce monotonic policy improvement to solve offline RL.

## 7 EXPERIMENTS

We conduct a series of experiments on the Gym (v2), Adroit (v1), Kitchen (v0) and Antmaze (v2) from D4RL (Fu et al., 2020) to evaluate the performance and analyze the design choice of **B**ehavior **P**roximal **P**olicy **O**ptimization (BPPO). Specifically, we aim to answer: 1) How does BPPO compare with previous Onestep and iterative methods? 2) What is the superiority of BPPO over its Onestep and iterative version? 3) What is the influence of hyperparameters clip ratio $\epsilon$ and clip ratio decay $\sigma$?

Table 1: The normalized results on D4RL Gym, Adroit, and Kitchen. We **bold** the best results and BPPO is calculated by averaging mean returns over 10 evaluation trajectories and five random seeds. The symbol * specifies that the results are reproduced by running the offical open-source code.

| Suite | Environment | Iterative methods | | Onestep methods | | BC (Ours) | BPPO (Ours) |
|---|---|---|---|---|---|---|---|
| | | CQL | TD3+BC | Onestep RL | IQL | | |
| Gym | halfcheetah-medium-v2 | 44.0 | **48.3** | **48.4** | 47.4 | 43.5±0.1 | 44.0±0.2 |
| | hopper-medium-v2 | 58.5 | 59.3 | 59.6 | 66.3 | 61.3±3.2 | **93.9±3.9** |
| | walker2d-medium-v2 | 72.5 | **83.7** | 81.8 | 78.3 | 74.2±4.6 | **83.6±0.9** |
| | halfcheetah-medium-replay-v2 | **45.5** | 44.6 | 38.1 | **44.2** | 40.1±0.1 | 41.0±0.6 |
| | hopper-medium-replay-v2 | 95.0 | 60.9 | **97.5** | 94.7 | 66.0±18.3 | 92.5±3.4 |
| | walker2d-medium-replay-v2 | 77.2 | **81.8** | 49.5 | 73.9 | 33.4±11.2 | 77.6±7.8 |
| | halfcheetah-medium-expert-v2 | 91.6 | 90.7 | **93.4** | 86.7 | 64.4±8.5 | **92.5±1.9** |
| | hopper-medium-expert-v2 | 105.4 | 98.0 | 103.3 | 91.5 | 64.9±7.7 | **112.8±1.7** |
| | walker2d-medium-expert-v2 | 108.8 | 110.1 | **113.0** | 109.6 | 107.7±3.5 | **113.1±2.4** |
| | *Gym locomotion-v2 total* | *698.5* | *677.4* | *684.6* | *692.4* | *555.5±57.2* | *751.0±21.8* |
| Adroit | pen-human-v1 | 37.5 | 8.4* | 90.7* | 71.5 | 61.6±9.7 | **117.8±11.9** |
| | hammer-human-v1 | 4.4 | 2.0* | 0.2* | 1.4 | 2.0±0.9 | **14.9±3.2** |
| | door-human-v1 | 9.9 | 0.5* | -0.1* | 4.3 | 7.8±3.5 | **25.9±7.5** |
| | relocate-human-v1 | 0.2 | -0.3* | 2.1* | 0.1 | 0.1±0.0 | **4.8±2.2** |
| | pen-cloned-v1 | 39.2 | 41.5* | 60.0 | 37.3 | 58.8±16.0 | **110.8±6.3** |
| | hammer-cloned-v1 | 2.1 | 0.8* | 2.0 | 2.1 | 0.5±0.2 | **8.9±5.1** |
| | door-cloned-v1 | 0.4 | -0.4* | 0.4 | 1.6 | 0.9±0.8 | **6.2±1.6** |
| | relocate-cloned-v1 | -0.1 | -0.3* | -0.1 | -0.2 | -0.1±0.0 | **1.9±1.0** |
| | *adroit-v1 total* | *93.6* | *52.2* | *155.2* | *118.1* | *131.6±31.1* | *291.4±38.8* |
| Kitchen | kitchen-complete-v0 | 43.8 | 0.0* | 2.0* | 62.5 | 55.0±11.5 | **91.5±8.9** |
| | kitchen-partial-v0 | 49.8 | 22.5* | 35.5* | 46.3 | 44.0±4.9 | **57.0±2.4** |
| | kitchen-mixed-v0 | 51.0 | 25.0* | 28.0* | 51.0 | 45.0±1.6 | **62.5±6.7** |
| | *kitchen-v0 total* | *144.6* | *47.5* | *65.5* | *159.8* | *144.0±18.0* | *211.0±18.0* |
| | *locomotion+kitchen+adroit* | *936.7* | *777.1* | *905.3* | *970.3* | *831.1±106.3* | *1253.4±78.6* |

## 7.1 RESULTS ON D4RL BENCHMARKS

We first compare BPPO with iterative methods including CQL (Kumar et al., 2020) and TD3+BC (Fujimoto & Gu, 2021), and Onestep methods including Onestep RL (Brandfonbrener et al., 2021) and IQL (Kostrikov et al., 2021). Most results of Onestep RL, IQL, CQL, TD3+BC are extracted from the paper IQL and the results with symbol * are reproduced by ourselves. Since BPPO first estimates a behavior policy and then improves it, we list the results of BC on the left side of BPPO.

From Table 1, we find BPPO achieves comparable performance on each task of Gym and slightly outperforms when considering the total performance. For Adroit and Kitchen, BPPO prominently outperforms other methods. Compared to BC, BPPO achieves 51% performance improvement on all D4RL tasks. Interestingly, our implemented BC on Adroit and Kitchen nearly outperform the baselines, which may imply improving behavior policy rather than learning from scratch is better.

Next, we evaluate whether BPPO can solve more difficult tasks with sparse reward. For Antmaze tasks, we also compare BPPO with Decision Transformer (DT) (Chen et al., 2021), RvS-G and RvS-R (Emmons et al., 2021). DT conditions on past trajectories to predict future actions using Transformer. RvS-G and RvS-R condition on goals or rewards to learn policy via supervised learning.

Table 2: The normalized results on D4RL Antmaze tasks. The results of CQL and IQL are extracted from paper IQL while others are extracted from paper RvS. In the BC column, symbol * specifies the Filtered BC (Emmons et al., 2021) which removes the failed trajectories instead of standard BC.

| Environment | CQL | TD3+BC | Onestep | IQL | DT | RvS-R | RvS-G | BC (Ours) | BPPO (Ours) |
|---|---|---|---|---|---|---|---|---|---|
| Umaze-v2 | 74.0 | 78.6 | 64.3 | 87.5 | 65.6 | 64.4 | 65.4 | 51.7±20.4 | **95.0±5.5** |
| Umaze-diverse-v2 | 84.0 | 71.4 | 60.7 | 62.2 | 51.2 | 70.1 | 60.9 | 48.3±17.2 | **91.7±4.1** |
| Medium-play-v2 | 61.2 | 10.6 | 0.3 | **71.2** | 1.0 | 4.5 | 58.1 | 16.7±5.2* | 51.7±7.5 |
| Medium-diverse-v2 | 53.7 | 3.0 | 0.0 | **70.0** | 0.6 | 7.7 | 67.3 | 33.3±10.3* | 70.0±6.3 |
| Large-play-v2 | 15.8 | 0.2 | 0.0 | 39.6 | 0.0 | 3.5 | 32.4 | 48.3±11.7* | **86.7±8.2** |
| Large-diverse-v2 | 14.9 | 0.0 | 0.0 | 47.5 | 0.2 | 3.7 | 36.9 | 46.7±20.7* | **88.3±4.1** |
| *Total* | *303.6* | *163.8* | *61.0* | *378.0* | *118.6* | *153.9* | *321.0* | *245.0±85.5* | ***483.3±35.7*** |

As shown in Table 2, BPPO can outperform most tasks and is significantly better than other algorithms in the total performance of all tasks. We adopt Filtered BC in last four tasks, where only the successful trajectories is selected for behavior cloning. The performance of CQL and IQL is very impressive since no additional operations or information is introduced. RvS-G uses the goal to overcome the sparse reward challenge. The superior performance demonstrates BPPO can also considerably improve the policy performance based on (Filtered) BC on tasks with sparse reward.

## 7.2 THE SUPERIORITY OF BPPO OVER ONESTEP AND ITERATIVE VERSION

**BPPO v.s. Onestep BPPO** We choose to improve policy $\pi_k$ after it has been improved over behavior policy $\hat{\pi}_\beta$ because Theorem 2 provides no guarantee of optimality. Besides, BPPO and Onestep RL are easily to be connected because BPPO is based on online method while Onestep RL solves offline RL without off-policy evaluation. Although Figure 2 gives an intuitive interpretation to show the advantage of BPPO over its Onestep version, the soundness is relatively weak. We further analyze the superiority of BPPO over its Onestep version empirically.

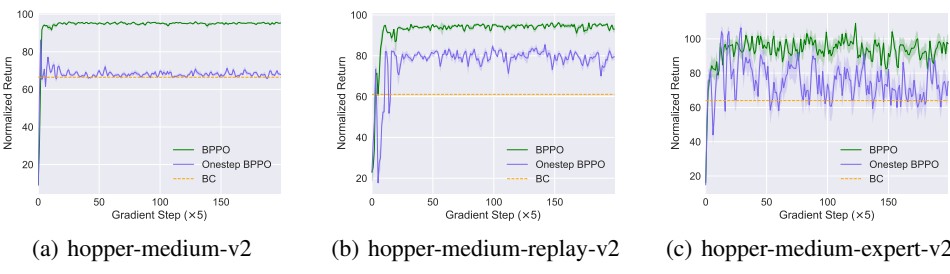

|  |  |  |
|---|---|---|
| (a) hopper-medium-v2 | (b) hopper-medium-replay-v2 | (c) hopper-medium-expert-v2 |

Figure 3: The comparison between BPPO and Onestep BPPO. The hyperparameters of both methods are tuned through the grid search, and then we exhibit their learning curves with the best performance.

In Figure3, we observe that both BPPO and Onestep BPPO can outperform BC (the orange dotted line). This indicates both of them can achieve monotonic improvement over behavior policy $\hat{\pi}_\beta$. Another important result is that BPPO is consitently better than Onestep BPPO and this demonstrates two key points: **First**, improving $\pi_k$ to fully utilize information is necessary. **Second**, compared to strictly restricting the learned policy close to the behavior policy, appropriate looseness is useful.

**BPPO v.s. iterative BPPO** When approximating the advantage $A_{\pi_k}$, we have two implementation choices. One is *advantage replacement* (line 11 in Algorithm 1). The other one is off-policy $Q$-estimation (line 13 in Algorithm 1), corresponding to iterative BPPO. Both of them will introduce extra error compared to true $A_{\pi_k}$. The error of the former comes from replacement $A_{\pi_k} \leftarrow A_{\pi_\beta}$ while the latter comes from the off-policy estimation itself. We compare BPPO with iterative BPPO in Figure 4 and find that *advantage replacement*, namely BPPO, is obviously better.

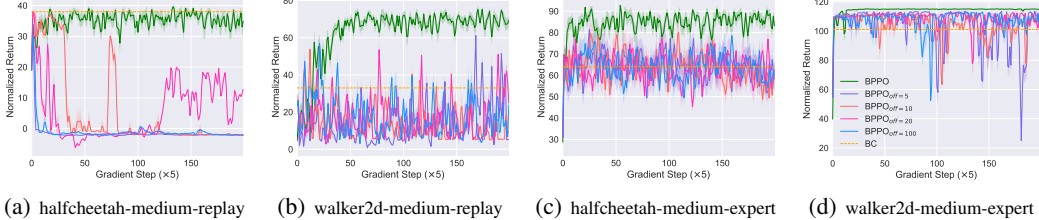

(a) halfcheetah-medium-replay (b) walker2d-medium-replay (c) halfcheetah-medium-expert (d) walker2d-medium-expert

Figure 4: The comparison between BPPO (the green curves) and its iterative versions in which we update the $Q$ network to approximate $Q_{\pi_k}$ instead of $Q_{\hat{\pi}_\beta}$ using in BPPO. In particular, we use "BPPO$_{off=5}$" to denote that we update $Q$ network for 5 gradient steps per policy training step.

## 7.3 ABLATION STUDY OF DIFFERENT HYPERPARAMETERS

In this section, we evaluate the influence of clip ratio $\epsilon$ and its decay rate $\sigma$. Clip ratio restricts the policy close to behavior policy and it directly solves the offline overestimation. Since $\epsilon$ also appears in PPO, we can set it properly to avoid catastrophic performance, which is the unique feature of BPPO. $\sigma$ gradually tightens this restriction during policy improvement. We show how these coefficients contribute to the performance of BPPO and more ablations can be found in Appendix G, I, and H.

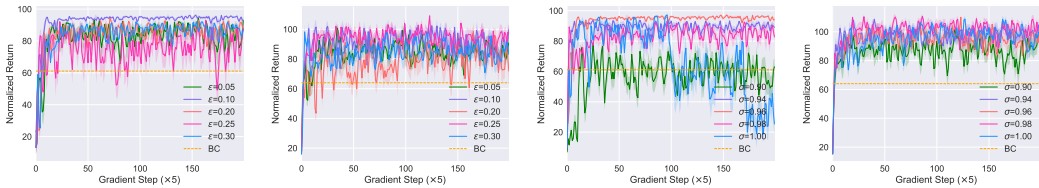

(a) hopper-medium-replay (b) hopper-medium-expert (c) hopper-medium-replay (d) hopper-medium-expert

Figure 5: Ablation study on clip ratio $\epsilon$ (5(a), 5(b)) and clip ratio decay $\sigma$ (5(c), 5(d)).

Firstly, we analyze five values of the clip coefficient $\epsilon = (0.05, 0.1, 0.2, 0.25, 0.3)$. In most environment, like *hopper-medium-expert* 5(b), different $\epsilon$ shows no significant difference so we choose $\epsilon = 0.25$, while only $\epsilon = 0.1$ is obviously better than others for *hopper-medium-replay*. We then demonstrate how the clip ratio decay ($\sigma = 0.90, 0.94, 0.96, 0.98, 1.00$) affects the performance of BPPO. As shown in Figure 5(c), a low decay rate ($\sigma = 0.90$) or no decay ($\sigma = 1.00$) may cause crash during training. We use $\sigma = 0.96$ to achieve stable policy improvement for all environments.

## 8 CONCLUSION

**B**ehavior **P**roximal **P**olicy **O**ptimization (BPPO) starts from offline monotonic policy improvement, using the loss function of PPO to elegantly solve offline RL without any extra constraint or regularization introduced. Theoretical derivations and extensive experiments show that the inherent conservatism from the on-policy method PPO is naturally suitable to overcome overestimation in offline RL. BPPO is simple to implement and achieves superior performance on the D4RL dataset.

## 9 ACKNOWLEDGEMENTS

This work was supported by the National Science and Technology Innovation 2030 - Major Project (Grant No. 2022ZD0208800), and NSFC General Program (Grant No. 62176215). We really appreciate Li He and Yachen Kang for helpful discussions and writing polishing.

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

## A   PROOF OF PERFORMANCE DIFFERENCE THEOREM 1

*Proof.* First note that $A_\pi(s, a) = \mathbb{E}_{s' \sim p(s'|s,a)} \left[ r(s, a) + \gamma V_\pi(s') - V_\pi(s) \right]$. Therefore,

$$
\begin{aligned}
&\mathbb{E}_{\tau \sim P_{\pi'}} \left[ \sum_{t=0}^{T} \gamma^t A_\pi(s_t, a_t) \right] \\
=&\mathbb{E}_{\tau \sim P_{\pi'}} \left[ \sum_{t=0}^{T} \gamma^t \left( r(s_t, a_t) + \gamma V_\pi(s_{t+1}) - V_\pi(s_t) \right) \right] \\
=&\mathbb{E}_{\tau \sim P_{\pi'}} \left[ -V_\pi(s_0) + \sum_{t=0}^{T} \gamma^t r(s_t, a_t) \right] \\
=& -\mathbb{E}_{s_0} \left[ V_\pi(s_0) \right] + \mathbb{E}_{\tau \sim P_{\pi'}} \left[ \sum_{t=0}^{T} \gamma^t r(s_t, a_t) \right] \\
=& -J(\pi) + J(\pi') \\
\triangleq& J_\Delta(\pi', \pi)
\end{aligned}
\tag{17}
$$

Now the first equation in 1 has been proved. For the proof of second equation, we decompose the expectation over the trajectory into the sum of expectation over state-action pairs:

$$
\begin{aligned}
&\mathbb{E}_{\tau \sim P_{\pi'}} \left[ \sum_{t=0}^{T} \gamma^t A_\pi(s_t, a_t) \right] \\
=& \sum_{t=0}^{T} \sum_{s} \left[ P(s_t = s|\pi') \mathbb{E}_{a \sim \pi'(\cdot|s)} \left[ \gamma^t A_\pi(s, a) \right] \right] \\
=& \sum_{s} \left[ \sum_{t=0}^{T} \gamma^t P(s_t = s|\pi') \mathbb{E}_{a \sim \pi'(\cdot|s)} \left[ A_\pi(s, a) \right] \right] \\
=& \sum_{s} \left[ \rho_{\pi'}(s) \mathbb{E}_{a \sim \pi'(\cdot|s)} \left[ A_\pi(s, a) \right] \right] \\
=& \mathbb{E}_{s \sim \rho_{\pi'}(s), a \sim \pi'(\cdot|s)} \left[ A_\pi(s, a) \right]
\end{aligned}
\tag{18}
$$

## B   PROOF OF PROPOSITION 1

*Proof.* For state-action pair $(s_t, a_t) \in \mathcal{D}$, it can be viewed as one deterministic policy that satisfies $\pi_\mathcal{D}(a = a_t|s_t) = 1$ and $\pi_\mathcal{D}(a \neq a_t|s_t) = 0$. So

$$
\begin{aligned}
&D_{TV}(\mathcal{D}\|\hat{\pi}_\beta)[s_t] = D_{TV}(\pi_\mathcal{D}\|\hat{\pi}_\beta)[s_t] \\
=& \frac{1}{2} \mathbb{E}_a |\pi_\mathcal{D}(a|s_t) - \hat{\pi}_\beta(a|s_t)| \\
=& \frac{1}{2} \int \left[ P(a_t) |\pi_\mathcal{D}(a_t|s_t) - \hat{\pi}_\beta(a_t|s_t)| + P(a \neq a_t) |\pi_\mathcal{D}(a|s_t) - \hat{\pi}_\beta(a|s_t)| \right] \mathrm{d}a \\
=& \frac{1}{2} \int \left[ P(a_t)(1 - \hat{\pi}_\beta(a_t|s_t)) + P(a \neq a_t) \hat{\pi}_\beta(a \neq a_t|s_t) \right] \mathrm{d}a \\
=& \frac{1}{2} \int \left[ P(a_t)(1 - \hat{\pi}_\beta(a_t|s_t)) + (1 - P(a_t))(1 - \hat{\pi}_\beta(a_t|s_t)) \right] \mathrm{d}a \\
=& \frac{1}{2}(1 - \hat{\pi}_\beta(a_t|s_t))
\end{aligned}
\tag{19}
$$

## C   PROOF OF THEOREM 2

The definition of $\bar{A}_{\pi,\hat{\pi}_\beta}(s)$ is as follows:

$$\bar{A}_{\pi,\hat{\pi}_\beta}(s) = \mathbb{E}_{a\sim\pi(\cdot|s)}\left[A_{\hat{\pi}_\beta}(s,a)\right] \tag{20}$$

Note that the expectation of advantage function $A_{\hat{\pi}_\beta}(s,a)$ depends on another policy $\pi$ rather than $\hat{\pi}_\beta$, so $\bar{A}_{\pi,\hat{\pi}_\beta}(s) \neq 0$. Furthermore, given the $\bar{A}_{\pi,\hat{\pi}_\beta}(s)$, the performance difference in Theorem 2 can be rewritten as:

$$J_\Delta(\pi,\hat{\pi}_\beta) = \mathbb{E}_{s\sim\rho_\pi(\cdot),a\sim\pi(\cdot|s)}\left[A_{\hat{\pi}_\beta}(s,a)\right] = \mathbb{E}_{s\sim\rho_\pi(\cdot)}\left[\bar{A}_{\pi,\hat{\pi}_\beta}(s)\right] \tag{21}$$

$$\widehat{J}_\Delta(\pi,\hat{\pi}_\beta) = \mathbb{E}_{s\sim\rho_\mathcal{D}(\cdot),a\sim\pi(\cdot|s)}\left[A_{\hat{\pi}_\beta}(s,a)\right] = \mathbb{E}_{s\sim\rho_\mathcal{D}(\cdot)}\left[\bar{A}_{\pi,\hat{\pi}_\beta}(s)\right] \tag{22}$$

**Lemma 1.** *For all state $s$,*

$$\left|\bar{A}_{\pi,\hat{\pi}_\beta}(s)\right| \leq 2\max_a\left|A_{\hat{\pi}_\beta}(s,a)\right| \cdot D_{TV}(\pi\|\hat{\pi}_\beta)[s] \tag{23}$$

*Proof.* The expectation of advantage function $A_\pi(s,a)$ over its policy $\pi$ equals zero:

$$\mathbb{E}_{a\sim\pi}\left[A_\pi(s,a)\right] = \mathbb{E}_{a\sim\pi}\left[Q_\pi(s,a) - V_\pi(s)\right] = \mathbb{E}_{a\sim\pi}\left[Q_\pi(s,a)\right] - V_\pi(s) = 0 \tag{24}$$

Thus, with the help of Hölder's inequality, we get

$$\begin{aligned}
\left|\bar{A}_{\pi,\hat{\pi}_\beta}(s)\right| &= \left|\mathbb{E}_{a\sim\pi(\cdot|s)}\left[A_{\hat{\pi}_\beta}(s,a)\right] - \mathbb{E}_{a\sim\hat{\pi}_\beta(\cdot|s)}\left[A_{\hat{\pi}_\beta}(s,a)\right]\right|\\
&\leq \|\pi(a|s) - \hat{\pi}_\beta(a|s)\|_1 \|A_{\hat{\pi}_\beta}(s,a)\|_\infty\\
&= 2D_{TV}(\pi\|\hat{\pi}_\beta)[s] \cdot \max_a\left|A_{\hat{\pi}_\beta}(s,a)\right|, \forall s
\end{aligned} \tag{25}$$

**Lemma 2.** ((Achiam et al., 2017)) *The divergence between two unnormalized visitation frequencies, $\|\rho_\pi(\cdot) - \rho_{\pi'}(\cdot)\|_1$, is bounded by an average total variational divergence of the policies $\pi$ and $\pi'$:*

$$\|\rho_\pi(\cdot) - \rho_{\pi'}(\cdot)\|_1 \leq 2\gamma\mathop{\mathbb{E}}_{s\sim\rho_{\pi'}(\cdot)}\left[D_{TV}(\pi\|\pi')[s]\right] \tag{26}$$

Given this powerful lemma and other preparation, now we are able to derive the bound of $\left|J_\Delta(\pi,\hat{\pi}_\beta) - \widehat{J}_\Delta(\pi,\hat{\pi}_\beta)\right|$:

$$\left|J_\Delta(\pi,\hat{\pi}_\beta) - \widehat{J}_\Delta(\pi,\hat{\pi}_\beta)\right| = \left|\mathbb{E}_{s\sim\rho_\pi(\cdot)}\left[\bar{A}_{\pi,\hat{\pi}_\beta}(s)\right] - \mathbb{E}_{s\sim\rho_\mathcal{D}(\cdot)}\left[\bar{A}_{\pi,\hat{\pi}_\beta}(s)\right]\right|$$

$$= \left|\left(\mathbb{E}_{s\sim\rho_\pi(\cdot)}\left[\bar{A}_{\pi,\hat{\pi}_\beta}(s)\right] - \mathbb{E}_{s\sim\rho_{\hat{\pi}_\beta}(\cdot)}\left[\bar{A}_{\pi,\hat{\pi}_\beta}(s)\right]\right) + \left(\mathbb{E}_{s\sim\rho_{\hat{\pi}_\beta}}\left[\bar{A}_{\pi,\hat{\pi}_\beta}(s)\right] - \mathbb{E}_{s\sim\rho_\mathcal{D}(\cdot)}\left[\bar{A}_{\pi,\hat{\pi}_\beta}(s)\right]\right)\right| \tag{27}$$

Based on Hölder's inequality and lemma 2, we can bound the first term as follows:

$$\left|\left(\mathbb{E}_{s\sim\rho_\pi(\cdot)}\left[\bar{A}_{\pi,\hat{\pi}_\beta}(s)\right] - \mathbb{E}_{s\sim\rho_{\hat{\pi}_\beta}(\cdot)}\left[\bar{A}_{\pi,\hat{\pi}_\beta}(s)\right]\right)\right| \leq \|\rho_\pi(\cdot) - \rho_{\hat{\pi}_\beta}(\cdot)\|_1 \|\bar{A}_{\pi,\hat{\pi}_\beta}(s)\|_\infty$$

$$\leq 2\gamma\mathop{\mathbb{E}}_{s\sim\rho_{\hat{\pi}_\beta}(\cdot)}\left[D_{TV}(\pi\|\hat{\pi}_\beta)[s]\right] \cdot \max_s\left|\bar{A}_{\pi,\hat{\pi}_\beta}(s)\right| \tag{28}$$

For the second term, we can derive similar bound and furthermore let $D_{TV}(\mathcal{D}\|\hat{\pi}_\beta)[s] = \frac{1}{2}(1 - \hat{\pi}_\beta(a_t|s_t))$. Finally, using lemma 1, we get

$$\left|J_\Delta(\pi,\hat{\pi}_\beta) - \widehat{J}_\Delta(\pi,\hat{\pi}_\beta)\right|$$

$$\leq 2\gamma\max_s\left|\bar{A}_{\pi,\hat{\pi}_\beta}(s)\right|\left(\mathop{\mathbb{E}}_{s\sim\rho_{\hat{\pi}_\beta}(\cdot)}\left[D_{TV}(\pi\|\hat{\pi}_\beta)[s]\right] + \mathop{\mathbb{E}}_{s\sim\rho_\mathcal{D}(\cdot)}\left[D_{TV}(\mathcal{D}\|\hat{\pi}_\beta)[s]\right]\right)$$

$$= 2\gamma\max_s\left|\bar{A}_{\pi,\hat{\pi}_\beta}(s)\right|\left(\mathop{\mathbb{E}}_{s\sim\rho_{\hat{\pi}_\beta}(\cdot)}\left[D_{TV}(\pi\|\hat{\pi}_\beta)[s]\right] + \mathop{\mathbb{E}}_{s\sim\rho_\mathcal{D}(\cdot)}\frac{1}{2}\left[1 - \hat{\pi}_\beta(a|s)\right]\right)$$

$$\leq 4\gamma\max_{s,a}\left|A_{\hat{\pi}_\beta}(s,a)\right| \cdot \max_s D_{TV}(\pi\|\hat{\pi}_\beta)[s] \cdot \left(\mathop{\mathbb{E}}_{s\sim\rho_{\hat{\pi}_\beta}(\cdot)}\left[D_{TV}(\pi\|\hat{\pi}_\beta)[s]\right] + \mathop{\mathbb{E}}_{s\sim\rho_\mathcal{D}(\cdot)}\frac{1}{2}\left[1 - \hat{\pi}_\beta(a|s)\right]\right) \tag{29}$$

# D  PROOF OF THEOREM 3

As an extension of Theorem 2, the proof process of Theorem 3 is similar. Based on the Equation (28), we can directly derive the final bound:

$$
\left| J_\Delta\left(\pi,\pi_k\right) - \widehat{J}_\Delta\left(\pi,\pi_k\right) \right| = \left| \mathbb{E}_{s\sim\rho_\pi(\cdot),a\sim\pi(\cdot|s)}\left[A_{\pi_k}(s,a)\right] - \mathbb{E}_{s\sim\rho_\mathcal{D}(\cdot),a\sim\pi(\cdot|s)}\left[A_{\pi_k}(s,a)\right] \right|
$$

$$
= \Bigg| \left( \mathbb{E}_{s\sim\rho_\pi(\cdot)}\left[\bar{A}_{\pi,\pi_k}(s)\right] - \mathbb{E}_{s\sim\rho_{\pi_k}(\cdot)}\left[\bar{A}_{\pi,\pi_k}(s)\right] \right) + \left( \mathbb{E}_{s\sim\rho_{\pi_k}(\cdot)}\left[\bar{A}_{\pi,\pi_k}(s)\right] - \mathbb{E}_{s\sim\rho_{\hat{\pi}_\beta}(\cdot)}\left[\bar{A}_{\pi,\pi_k}(s)\right] \right)
$$

$$
+ \left( \mathbb{E}_{s\sim\rho_{\hat{\pi}_\beta}(\cdot)}\left[\bar{A}_{\pi,\pi_k}(s)\right] - \mathbb{E}_{s\sim\rho_\mathcal{D}(\cdot)}\left[\bar{A}_{\pi,\pi_k}(s)\right] \right) \Bigg|
$$

$$
\leq 2\gamma \max_s \left|\bar{A}_{\pi,\pi_k}(s)\right| \left( \mathbb{E}_{s\sim\rho_{\pi_k}(\cdot)}\left[D_{TV}\left(\pi\|\pi_k\right)[s]\right] + \mathbb{E}_{s\sim\rho_{\hat{\pi}_\beta}(\cdot)}\left[D_{TV}\left(\pi_k\|\hat{\pi}_\beta\right)[s]\right] + \mathbb{E}_{s\sim\rho_\mathcal{D}(\cdot)}\frac{1}{2}\left[1-\hat{\pi}_\beta\left(a|s\right)\right] \right)
$$

$$
\leq 4\gamma \max_{s,a} \left|A_{\pi_k}(s,a)\right| \cdot \max_s D_{TV}\left(\pi\|\pi_k\right)[s]
$$

$$
\cdot \left( \mathbb{E}_{s\sim\rho_{\pi_k}(\cdot)}\left[D_{TV}\left(\pi\|\pi_k\right)[s]\right] + \mathbb{E}_{s\sim\rho_{\hat{\pi}_\beta}(\cdot)}\left[D_{TV}\left(\pi_k\|\hat{\pi}_\beta\right)[s]\right] + \mathbb{E}_{s\sim\rho_\mathcal{D}(\cdot)}\frac{1}{2}\left[1-\hat{\pi}_\beta\left(a|s\right)\right] \right)
$$

$$
\tag{30}
$$

# E  WHY GAE IS UNAVAILABLE IN OFFLINE SETTING?

In traditional online situation, advantage $A_{\pi_k}(s,a)$ is estimated by Generalized Advantage Estimation (GAE) (Schulman et al., 2015b) using the data collected by policy $\pi_k$. But in offline RL, only offline dataset $\mathcal{D} = \left\{ (s_t, a_t, s_{t+1}, r_t)_{t=1}^N \right\}$ from true behavior policy $\pi_\beta$ is available. The advantage of $(s_t, a_t)$ calculated by GAE is as follow:

$$
A_{\pi_\beta}\left(s_t, a_t\right) = \sum_{l=0}^\infty (\gamma\lambda)^l \left( r_{t+l} + \gamma V_{\pi_\beta}\left(s_{t+l+1}\right) - V_{\pi_\beta}\left(s_{t+l}\right) \right). \tag{31}
$$

GAE can only calculate the advantage of $(s_t, a_t) \in \mathcal{D}$. For $(s_t, \tilde{a}_t) \sim \mathcal{D}$, where $\tilde{a}_t$ is an in-distribution action sampling but $(s_t, \tilde{a}_t) \notin \mathcal{D}$, GAE is **unable** to give any estimation. This is because the calculation process of GAE depends on the trajectory and does not have the ability to generalize to unseen state-action pairs. Therefore, GAE is not a satisfactory choice for offline RL. Offline RL forbids the interaction with environment, so data usage should be more efficient. Concretely, we expect advantage approximation method can not only calculate the advantage of $(s_t, a_t)$, but also $(s_t, \tilde{a}_t)$. As a result, we directly estimate advantage with the definition $A_{\pi_\beta}(s,a) = Q_{\pi_\beta}(s,a) - V_{\pi_\beta}(s)$, where $Q$-function is estimated by SARSA and value function by fitting returns $\sum_{t=0}^T \gamma^t r(s_t, a_t)$ with MSE loss. This function approximation method can generalize to the advantage of $(s_t, \tilde{a}_t)$.

# F  THEORETICAL ANALYSIS FOR *Advantage Replacement*

We choose to replace all $A_{\pi_k}$ with trustworthy $A_{\hat{\pi}_\beta}$ then theoretically measure the difference rather than empirically make $A_{\pi_k}$ learned by Q-learning more accurate. The difference caused by replacing the $A_{\pi_k}$ in $\widehat{J}_\Delta\left(\pi,\pi_k\right)$ with $A_{\pi_\beta}(s,a)$ can be measured in the following theorem:

**Theorem 4.** *Given the distance $D_{TV}\left(\pi_k\|\pi_\beta\right)[s]$ and assume the reward function satisfies $|r(s,a)| \leq R_{max}$ for all $s, a$, then*

$$
\left| \widehat{J}_\Delta\left(\pi,\pi_k\right) - \mathbb{E}_{s\sim\rho_\mathcal{D}(\cdot),a\sim\pi(\cdot|s)}\left[A_{\pi_\beta}(s,a)\right] \right| \leq 2\gamma(\gamma+1)\cdot R_{max}\cdot \mathbb{E}_{s\sim\rho_{\pi_\beta}(\cdot)}\left[D_{TV}\left(\pi_k\|\pi_\beta\right)[s]\right].
$$

$$
\tag{32}
$$

*Proof.* First note that $A_\pi(s, a) = \mathbb{E}_{s' \sim p(s'|s,a)}[r(s, a) + \gamma V_\pi(s') - V_\pi(s)]$. Then we have

$$\left| \mathbb{E}_{s \sim \rho_\mathcal{D}(\cdot), a \sim \pi(\cdot|s)} \left[ A_{\pi_k}(s, a) \right] - \mathbb{E}_{s \sim \rho_\mathcal{D}(\cdot), a \sim \pi(\cdot|s)} \left[ A_{\pi_\beta}(s, a) \right] \right|$$

$$= \left| \mathbb{E}_{s \sim \rho_\mathcal{D}(\cdot), a \sim \pi(\cdot|s)} \mathbb{E}_{s' \sim p(s'|s,a)} \left[ \gamma \left( V_{\pi_k}(s') - V_{\pi_\beta}(s') \right) - \left( V_{\pi_k}(s) - V_{\pi_\beta}(s) \right) \right] \right|$$

$$\leq \mathbb{E}_{s \sim \rho_\mathcal{D}(\cdot), a \sim \pi(\cdot|s)} \mathbb{E}_{s' \sim p(s'|s,a)} \left[ \gamma \left| V_{\pi_k}(s') - V_{\pi_\beta}(s') \right| + \left| V_{\pi_k}(s) - V_{\pi_\beta}(s) \right| \right] \quad (33)$$

Similarly to Equation (18), the value function can be rewritten as $V_\pi(s) = \mathbb{E}_{s \sim \rho_\pi(\cdot)}[r(s)]$. Then the difference between two value function can be measured using Hölder's inequality and lemma 2:

$$\left| V_{\pi_k}(s) - V_{\pi_\beta}(s) \right| = \left| \mathbb{E}_{s \sim \rho_{\pi_k}(\cdot)}[r(s)] - \mathbb{E}_{s \sim \rho_{\pi_\beta}(\cdot)}[r(s)] \right|$$

$$\leq \left\| \rho_{\pi_k}(\cdot) - \rho_{\pi_\beta}(\cdot) \right\|_1 \left\| r(s) \right\|_\infty \leq 2\gamma \mathbb{E}_{s \sim \rho_{\pi_\beta}(\cdot)} \left[ D_{TV}(\pi_k \| \pi_\beta)[s] \right] \cdot \max_s |r(s)| \quad (34)$$

Thus, the final bound is

$$\left| \mathbb{E}_{s \sim \rho_\mathcal{D}(\cdot), a \sim \pi(\cdot|s)} \left[ A_{\pi_k}(s, a) \right] - \mathbb{E}_{s \sim \rho_\mathcal{D}(\cdot), a \sim \pi(\cdot|s)} \left[ A_{\pi_\beta}(s, a) \right] \right|$$

$$\leq \mathbb{E}_{s \sim \rho_\mathcal{D}(\cdot), a \sim \pi(\cdot|s)} \mathbb{E}_{s' \sim p(s'|s,a)} \left[ 2\gamma^2 \mathbb{E}_{s' \sim \rho_{\pi_\beta}(\cdot)} \left[ D_{TV}(\pi_k \| \pi_\beta)[s'] \right] \cdot \max_{s'} |r(s')| \right.$$

$$\left. + 2\gamma \mathbb{E}_{s \sim \rho_{\pi_\beta}(\cdot)} \left[ D_{TV}(\pi_k \| \pi_\beta)[s] \right] \cdot \max_s |r(s)| \right]$$

$$= 2\gamma(\gamma + 1) \max_s |r(s)| \mathbb{E}_{s \sim \rho_{\pi_\beta}(\cdot)} \left[ D_{TV}(\pi_k \| \pi_\beta)[s] \right] \quad (35)$$

Note that the right end term of the equation is irrelevant to the policy $\pi$ and can be viewed as a constant when optimizing $\pi$. Combining the result of Theorem 3 and 4, we get the following corollary:

**Corollary 1.** *Given the distance $D_{TV}(\pi \| \pi_k)[s]$, $D_{TV}(\pi_k \| \hat{\pi}_\beta)[s]$ and $D_{TV}(\mathcal{D} \| \hat{\pi}_\beta)[s] = \frac{1}{2}(1 - \hat{\pi}_\beta(a|s))$, we can derive the following bound:*

$$J_\Delta(\pi, \pi_k) \geq \mathbb{E}_{s \sim \rho_\mathcal{D}(\cdot), a \sim \pi(\cdot|s)} \left[ A_{\pi_\beta}(s, a) \right]$$

$$- 4\gamma \mathbb{A}_{\pi_k} \cdot \max_s D_{TV}(\pi \| \pi_k)[s] \cdot \mathbb{E}_{s \sim \rho_{\pi_k}(\cdot)} \left[ D_{TV}(\pi \| \pi_k)[s] \right]$$

$$- 4\gamma \mathbb{A}_{\pi_k} \cdot \max_s D_{TV}(\pi \| \pi_k)[s] \cdot \mathbb{E}_{s \sim \rho_{\hat{\pi}_\beta}(\cdot)} \left[ D_{TV}(\pi_k \| \hat{\pi}_\beta)[s] \right]$$

$$- 2\gamma \mathbb{A}_{\pi_k} \cdot \max_s D_{TV}(\pi \| \pi_k)[s] \cdot \mathbb{E}_{s \sim \rho_\mathcal{D}(\cdot)} \left[ 1 - \hat{\pi}_\beta(a|s) \right] - \boxed{\mathcal{C}_{\pi_k, \pi_\beta}}, \quad (36)$$

*where $\mathbb{A}_{\pi_k} = \max_{s,a} |A_{\pi_k}(s, a)|$ and $\mathcal{C}_{\pi_k, \pi_\beta} = 2\gamma(\gamma + 1) \cdot \max_{s,a} |r(s, a)| \mathbb{E}_{s \sim \rho_{\pi_\beta}(\cdot)} \left[ D_{TV}(\pi_k \| \pi_\beta)[s] \right]$.*

---

**Conclusion 3**

To guarantee the true objective $J_\Delta(\pi, \pi_k)$ non-decreasing, we can also simultaneously maximize $\mathbb{E}_{s \sim \rho_\mathcal{D}(\cdot), a \sim \pi(\cdot|s)} \left[ A_{\pi_\beta}(s, a) \right]$ and minimize $[\max_s D_{TV}(\pi \| \pi_k)[s]]$, $k = 0, 1, 2, \cdots$.

---

# G ABLATION STUDY ON AN ASYMMETRIC COEFFICIENT

In this section, we give the details of all hyperparameter selections in our experiments. In addition to the aforementioned clip ratio $\epsilon$ and its clip decay coefficient $\sigma$, we introduce the $\omega \in (0, 1)$ as an asymmetric coefficient to adjust the advantage $\bar{A}_{\pi_\beta}$ based on the positive or negative of advantage:

$$\bar{A}_{\pi_\beta} = |\omega - \mathbf{1}(A_{\pi_\beta} < 0)| A_{\pi_\beta}. \quad (37)$$

For $\omega > 0.5$, that downweights the contributions of the state-action value $Q_{\pi_\beta}$ smaller than it's expectation, i.e., $V_{\pi_\beta}$ while distributing more weights to larger $Q_{\pi_\beta}$. The asymmetric coefficient can adjust the weight of advantage based on the $Q$ performance, which downweights the contributions of the state-action value $Q$ smaller than its expectation while distributing more weights to advantage with a larger $Q$ value. We analyze how the three coefficients affect the performance of BPPO.

We analyze three values of the asymmetric coefficient $\omega = (0.5, 0.7, 0.9)$ in three Gym environments. Figure 6 shows that $\omega = 0.9$ is best for these tasks, especially in *hopper-medium-v2* and *hopper-medium-replay-v2*. With a larger value $\omega$, the policy improvement can be guided in a better direction, leading to better performance in Gym environments. Based on the performance of different coefficient values above, we use the asymmetric advantage coefficient $\omega = 0.9$ for the Gym dataset training and $\omega = 0.7$ for the Adroit, Antmaze, and Kitchen datasets training, respectively.

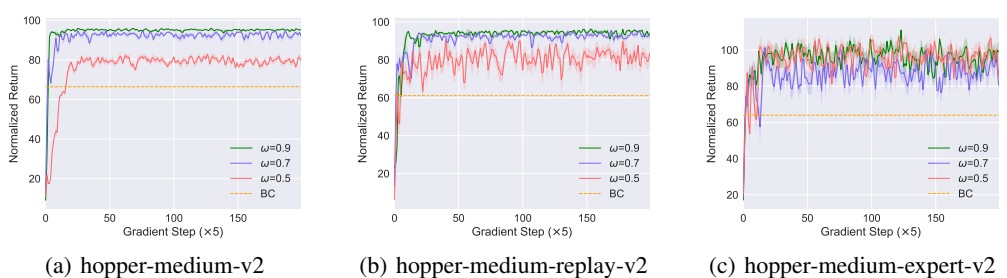

(a) hopper-medium-v2    (b) hopper-medium-replay-v2    (c) hopper-medium-expert-v2

Figure 6: Ablation study on coefficient $\omega$. We optimize the hyperparameters through the grid search, then we fix the value of other coefficients with the best performance and change the value of the asymmetric coefficient to analyze how it affects the BPPO. In particular, $\omega = 0.5$ denotes without the asymmetric coefficient during the training phase (contributing equal value to all Advantages).

## H  IMPORTANCE RATIO DURING TRAINING

In this section, we consider exploring whether the importance weight between the improved policy $\pi_k$ and the behavior policy $\pi_\beta$ will be arbitrarily large. To this end, we quantify this importance weight in the training phase in Figure 7. In Figure 7, we often observe that the ratio of the BPPO with decay always stays in the clipped region (the region surrounded by the dotted yellow and red line). However, the BPPO without decay is beyond the region in Figure 7(a) and 7(b). That demonstrates the improved policy without decay is farther away from the behavior policy than the case of BPPO with decay. It may cause unstable performance and even crashing, as shown in Figure 5(c), 5(d) and 10 when $\sigma = 1.00$ (i.e., without decay).

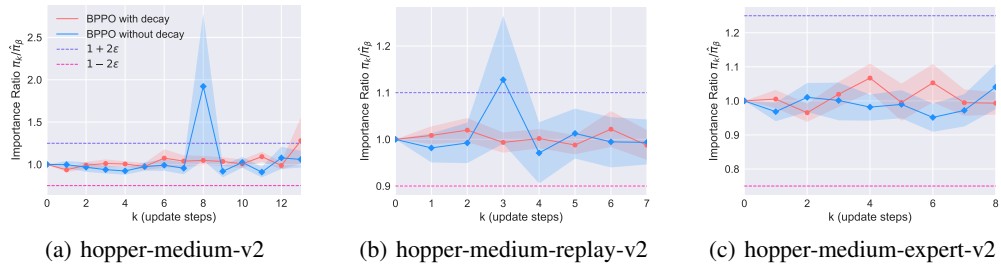

(a) hopper-medium-v2    (b) hopper-medium-replay-v2    (c) hopper-medium-expert-v2

Figure 7: Visualization of the importance weight between the updated policy and the behavior policy trained by BC. When the performance of the policy is improved, we calculate the importance weight (i.e., the probability ratio) between the improved policy and the behavior policy.

# I  COEFFICIENT PLOTS OF ONESTEP BPPO

In this section, we exhibit the learning curves and coefficient plots of Onestep BPPO. As shown in Figure 8 and 9, $\epsilon = 0.25$ and $\omega = 0.9$ are best for those tasks. Figure 10 shows how the clip coefficient decay affects the performance of the Onestep BPPO. We can observe that the performance of the curve without decay or with low decay is unstable over three tasks and even crash during training in the *"hopper-medium-replay-v2"* task. Thus, we select $\sigma = 0.96$ to achieve a stable policy improvement for Onestep BPPO. that We use the coefficients with the best performance to compare with the BPPO in Figure 3.

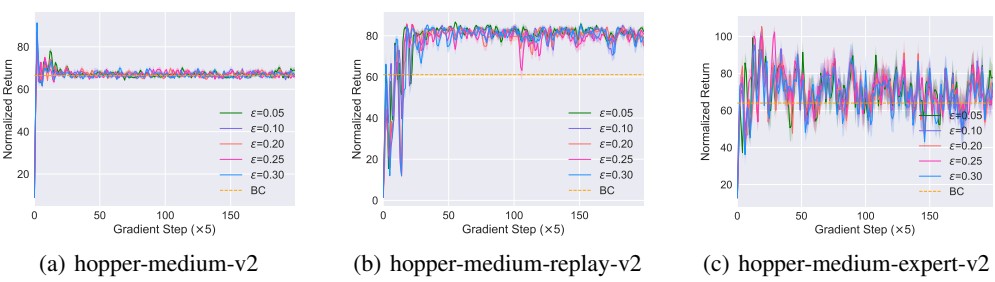

Figure 8: Ablation study of Onestep BPPO on coefficient $\epsilon$. We optimize the hyperparameters through the grid search, then we fix the value of other coefficients with the best performance and change the value of the clip coefficient to analyze how it affects the Onestep BPPO.

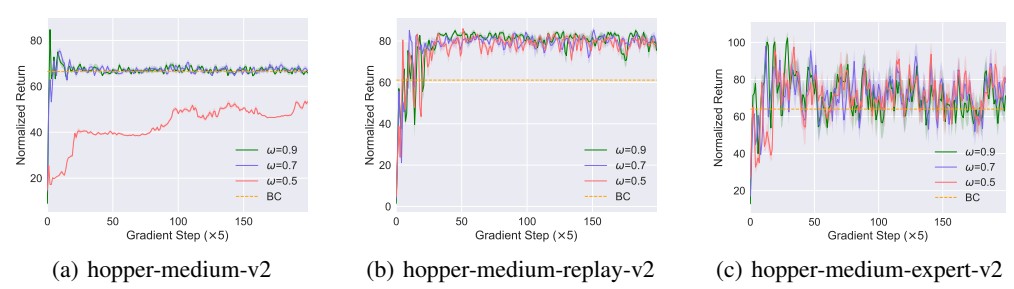

Figure 9: Ablation study of Onestep BPPO on coefficient $\omega$. We optimize the hyperparameters through the grid search, then we fix the value of other coefficients with the best performance and change the value of the asymmetric coefficient to analyze how it affects the Onestep BPPO.

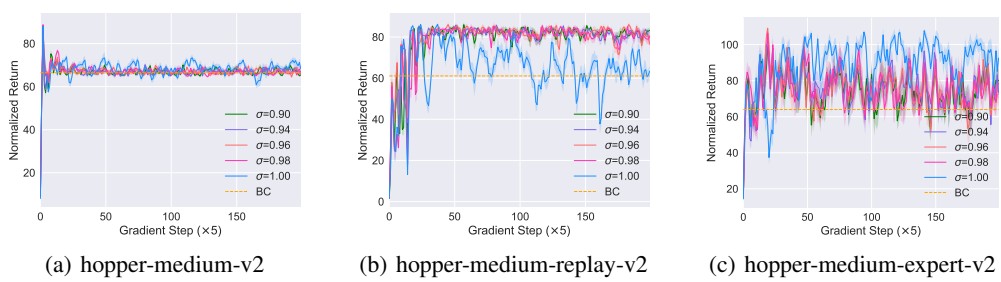

Figure 10: Ablation study of Onestep BPPO on clip coefficient decay and its decay rate. We optimize the hyperparameters through the grid search, then we fix the value of other coefficients with the best performance and change the value of the clip decay coefficient to analyze how it affects the Onestep BPPO. In particular, $\sigma = 1.00$ denotes without the decay coefficient during the training phase.

## J    EXTRA COMPARISONS

In this section, we have added the EDAC (An et al., 2021a), LAPO (Chen et al., 2022), RORL (Yang et al., 2022), and ATAC (Cheng et al., 2022) as the comparison baselines to further evaluate the superiority of the BPPO. Although the performance of the BPPO is slightly worse than the SOTA methods on Gym environment, the BPPO significantly outperforms all methods on the Adroit, Kitchen, and Antmaze datasets and has the best overall performance over all datasets.

Table 3: The normalized results of all algorithms on Gym locomotion and Adroit datasets. The results of the EDAC, RORL, and ATAC are extracted from their original articles.

| Environment/method | EDAC | RORL | ATAC | Ours |
|---|---|---|---|---|
| halfcheetah-medium-v2 | **65.9** | **66.8** | 54.3 | 44.0±0.2 |
| hopper-medium-v2 | 101.6 | **104.8** | 102.8 | 93.9±3.9 |
| walker2d-medium-v2 | 92.5 | **102.4** | 91.0 | 83.6±0.9 |
| halfcheetah-medium-replay-v2 | 61.3 | 61.9 | 49.5 | 41.0±0.6 |
| hopper-medium-replay-v2 | 101 | **102.8** | **102.8** | 92.5±3.4 |
| walker2d-medium-replay-v2 | 87.1 | 90.4 | **94.1** | 77.6±7.8 |
| halfcheetah-medium-expert-v2 | **106.3** | 107.8 | 95.5 | 92.5±1.9 |
| hopper-medium-expert-v2 | 110.7 | **112.7** | 112.6 | **112.8±1.7** |
| walker2d-medium-expert-v2 | 114.7 | 121.2 | **116.3** | 113.1±2.4 |
| *Gym locomotion-v2 total* | *841.1* | *870.8* | *818.9* | *751.0±21.8* |
| pen-human-v1 | 52.1 | 33.7 | 79.3 | **117.8±11.9** |
| hammer-human-v1 | 0.8 | 2.3 | 6.7 | **14.9±3.2** |
| door-human-v1 | 10.7 | 3.8 | 8.7 | **25.9±7.5** |
| relocate-human-v1 | 0.1 | 0 | 0.3 | **4.8±2.2** |
| pen-cloned-v1 | 68.2 | 35.7 | 73.9 | **110.8±6.3** |
| hammer-cloned-v1 | 0.3 | 1.7 | 2.3 | **8.9±5.1** |
| door-cloned-v1 | **9.6** | -0.1 | 8.2 | 6.2±1.6 |
| relocate-cloned-v1 | 0 | 0 | 0.8 | **1.9±1.0** |
| *adroit-v1 total* | *141.8* | *77.1* | *180.2* | ***291.4±38.8*** |
| *locomotion + adroit total* | *982.9* | *947.9* | *999.1* | ***1042.4±60.6*** |

Table 4: The normalized results of all algorithms on Kitchen dataset. The results of the LAPO are extracted from its original article.

| Environment/method | LAPO | Ours |
|---|---|---|
| kitchen-complete-v0 | 53.2 | **91.5±8.9** |
| kitchen-partial-v0 | 53.7 | **57.0±2.4** |
| kitchen-mixed-v0 | **62.4** | **62.5±6.7** |
| *kitchen-v0 total* | *169.3* | ***211.0±18.0*** |

Table 5: The normalized results of all algorithms on Antmaze dataset. The results of the RORL are extracted from its original article.

| Environment/method | RORL | Ours |
|---|---|---|
| Umaze-v2 | **96.7** | **95.0±5.5** |
| Umaze-diverse-v2 | **90.7** | **91.7±4.1** |
| Medium-play-v2 | **76.3** | 51.7±7.5 |
| Medium-diverse-v2 | **69.3** | **70.0±6.3** |
| Large-play-v2 | 16.3 | **86.7±8.2** |
| Large-diverse-v2 | 41.0 | **88.3±4.1** |
| *Antmaze-v2 total* | *390.3* | ***483.3±35.7*** |

## K    IMPLEMENTATION AND EXPERIMENT DETAILS

Following the online PPO method, we use tricks called 'code-level optimization' including learning rate decay, orthogonal initialization, and normalization of the advantage in each mini-batch, which are considered very important to the success of the online PPO algorithm (Engstrom et al., 2020). We clip the concatenated gradient of all parameters such that the 'global L2 norm' does not exceed 0.5. We use 2 layers MLP with 1024 hidden units for the $Q$ and policy networks, and use 3 layers MLP with 512 hidden units for value function $V$. Our method is constructed by Pytorch (Paszke et al., 2019). Next, we introduce the training details of the $Q, V$, (estimated) behavior policy $\hat{\pi}_\beta$, and target policy $\pi$, respectively.

- $Q$ and $V$ networks training: we run $2 \times 10^6$ steps for fitting value $Q$ and $V$ functions using learning rate $10^{-4}$, respectively.

- (Estimated) behavior policy $\hat{\pi}_\beta$ training: we run $5 \times 10^5$ steps for $\hat{\pi}_\beta$ cloning using learning rate $10^{-4}$.

- Target policy $\pi$ training: during policy improvement, we use the learning rate decay, i.e., decaying in each interval step in the first 200 gradient steps and then remaining the learning rate (decay rate $\sigma = 0.96$). We run 1,000 gradient steps for policy improvement for Gym, Adroit, and Kitchen tasks and run 100 gradient steps for Antmaze tasks. The selections of the initial policy learning rate, initial clip ratio, and asymmetric coefficient are listed in Table 6, respectively.

Table 6: The selections of part of hyperparameters during policy improvement phase.

| Hyperparameter | Task | Value |
|---|---|---|
| Initial policy learning rate | Gym locomotion and cloned tasks of Adroit | $10^{-4}$ |
| | Kitchen, Antmaze, and human tasks of Adroit | $10^{-5}$ |
| Initial clip ratio $\epsilon$ | Hopper-medium-replay-v2 | 0.1 |
| | Antmaze | 0.5 |
| | Others | 0.25 |
| Asymmetric coefficient $\omega$ | Gym locomotion | 0.9 |
| | Others | 0.7 |

