# OpenReview forum: "Behavior Proximal Policy Optimization "
_ICLR.cc/2023/Conference — ICLR 2023 poster_

### Official Review · Reviewer_ksVK · 2022-10-24

**Confidence:** 3
**Clarity, Quality, Novelty And Reproducibility:** Overall speaking, the writing the pap…
**Correctness:** 3
**Technical Novelty And Significance:** 2
**Empirical Novelty And Significance:** 2
**Recommendation:** 5

**Strength And Weaknesses:**

Strength: This paper provides theoretical analysis for monotonic policy improvement of the BPPO. Empirically, BPPO works for the vast majority of D4RL dataset.

Weakness:

1. Assumption 1 seems a little strong. Assuming $\hat{\pi}_\beta=\pi_\beta$ is not very reasonable since $\hat{\pi}_\beta$ is learned from the data. Ideally, there should be an expression that bounds the difference between $\pi$ and $\hat{\pi}$ using data.

2. equation (4), it is not reasonable to directly replace $\rho_\pi$ with state data distribution since the main challenge of offline is the data shift. As a result, there should be an importance ratio in front of it and should be estimated. Simply replacing the expression is not principled.

3. While policy improvement guarantee is nice, it cannot be guaranteed to converge to the optimal policy. Can you further show that your BPPO converge to the optimal policy after $K$ iterations?

4. Empirically, how will BPPO work for the sparse reward setting? one recent offline RL works all tackle the Antmaze environment as well.


**Summary Of The Paper:**

In this work, starting from the analysis of offline monotonic policy improvement, this paper gets a surprising finding that some online on-policy algorithms are naturally able to solve offline RL. Specifically, the inherent conservatism of these on-policy algorithms is exactly what the offline RL method needs to accomplish the closeness. Based on this, this paper designs an algorithm called Behavior Proximal Policy Optimization (BPPO), which successfully solves offline RL without any extra constraint or regularization introduced. Extensive experiments on the D4RL benchmark indicate this extremely succinct method outperforms state-of-the-art offline RL algorithms.


**Summary Of The Review:**

Please answer my question in above.

---

> ### Author Response · Authors · 2022-11-17
> **## Response to Reviewer ksVK**
>
> Thank reviewer for the approval. We have answered the theoretical concerns (**A1**, **A2**, **A3**) and experimental concerns (**A4**) in detail.
>
> **Q1: Assumption 1 is too strong.**
>
> **A1:** Assumption 1 is too strong to achieve in practice. In the latest submission version, we replace this assumption with the distance between estimated behavior policy and offline dataset using total variational divergence (refer to proposition 1). This distance only relates to the offline dataset, namely "bounds the difference using data". Besides, this modification does not change the final theoretical results.
>
> **Q2: Is it reasonable to directly replace $\rho_{\pi}$ with state distribution of offline dataset?**
>
> **A2:** After directly replacing $\rho_{\pi}$ with state distribution of offline dataset, we measure the difference by theorem2. The replacement is utilized by online monotonic policy improvement [1] and we adopt the similar manner. As for the "data shift" in offline RL, it represents the phenomenon that the state distribution of $\pi$ (the policy to be improved) mismatches the state distribution from offline dataset $\mathcal{D}$ (collected by this behavior policy $\pi_{\beta}$). Our method sets the policy to be improved as $\hat\pi_{\beta}$ estimated by the behavior cloning using $\mathcal{D}$, which just reduces the data shift. Besides directly replacing the state distribution then theoretically measuring, the importance sampling over state distribution also works [2]. To maintain the continuity of monotonic policy improvement theory, we directly replace the $\rho_{\pi}$ and this avoids the complex approximation of the importance ratio.
>
> **Q3: Can BPPO converge to the optimal policy?**
>
> **A3:** All the monotonic policy improvement theory does lack the optimal guarantee, but we can ceaselessly improve the current policy until the performance cannot be improved. It is intuitively reasonable to regard the policy obtained by this way as optimal policy. In the theory of offline monotonic policy improvement, we first derive how to monotonically improve the estimated behavior policy $\hat\pi_{\beta}$ then analyze how to improve the policy $\pi_k$, one policy has been improved over $\hat\pi_{\beta}$. By doing this, optimal policy can be indirectly attained by ceaselessly improving over $\hat\pi_{\beta}$.
>
> **Q4: Empirically, how will BPPO work for the sparse reward setting? one recent offline RL works all tackle the Antmaze environment as well.**
>
> **A4:** We have conducted a series of experiments to evaluate the performance of the BPPO on the Antmaze environments with sparse reward. Meanwhile, we compare the results of the BPPO with the state-of-the-art offline RL works in Section 6.2 in latest submission. The comparison shows that the BPPO can often outperform those works. Please see the latest submission for more details.
>
> | Environment       | TD3+BC  | Onestep | CQL     | IQL      | DT      | RvS-R   | RvS-G   | BC (Ours) | BPPO (Ours) |
> | ----------------- | ------- | ------- | ------- | -------- | ------- | ------- | ------- | --------- | ----------- |
> | Umaze-v2          | 78.6    | 64. 3   | 74.0    | 87.5     | 65.6    | 64.4    | 65.4    | 51.7      | **95.0**    |
> | Umaze-diverse-v2  | 71.4    | 60.7    | 84.0    | 62.2     | 51.2    | 70.1    | 60.9    | 48.3      | **91.7**    |
> | Medium-play-v2    | 10.6    | 0.3     | 61.2    | **71.2** | 1.0     | 4.5     | 58.1    | 16.7      | 51.7        |
> | Medium-diverse-v2 | 3.0     | 0.0     | 53.7    | **70.0** | 0.6     | 7.7     | 67.3    | 33.3      | **70.0**    |
> | Large-play-v2     | 0.2     | 0.0     | 15.8    | 39.6     | 0.0     | 3.5     | 32.4    | 48.3      | **86.7**    |
> | Large-diverse-v2  | 0.0     | 0.0     | 14.9    | 47.5     | 0.2     | 3.7     | 36.9    | 46.7      | **88.3**    |
> | *Total*           | *163.8* | *61.0*  | *303.6* | *378.0*  | *118.6* | *153.9* | *321.0* | *245.0*   | ***483.3*** |
>
> [1] Schulman J, Levine S, Abbeel P, et al. Trust region policy optimization[C]//International conference on machine learning. PMLR, 2015: 1889-1897.
>
> [2] Xu H, Zhan X, Li J, et al. Offline reinforcement learning with soft behavior regularization[J]. arXiv preprint arXiv:2110.07395, 2021.
>
> Thank you for your review again. We hope we have resolved your concerns. We are always willing to answer any of your further concerns.

---

> ### Author Response · Authors · 2022-12-10
> **Response to Reviewer ksVK during rebuttal**
>
> Here is a kind reminder that the Reviewer-Author discussion phase is coming to an end. We'd like to know whether our response has answered your question and we are always looking forward to your further comments or suggestions. Hope to hear you soon.

---

### Official Review · Reviewer_QTZM · 2022-10-24

**Confidence:** 3
**Correctness:** 3
**Technical Novelty And Significance:** 2
**Empirical Novelty And Significance:** 1
**Recommendation:** 6

**Clarity, Quality, Novelty And Reproducibility:**

The paper is clear to read, easy to follow and understand. The work is reasonably high quality combining an interesting theoretical insight with practical execution through a simple algorithm. The work is not necessarily novel, as prior approaches like Onestep RL are very similar in principle to the proposed algorithm. I haven't carefully looked at the proofs of the theorems.

**Strength And Weaknesses:**

Strengths

- The proposed algorithm is simple to implement and is only a slight modification to existing algorithm PPO.
- The insight of conservatism naturally emerging from online on-policy RL is neat, and interesting.
- The experiments are on a standard benchmark which is good from the perspective of head-to-head comparison and reproducibility

Weaknesses

- Assumption 1 is likely to be violated in several practical settings if the value of $\zeta$ is low. The experiments section doesnt provide any insight on the feasibility of this assumption while learning the behavior policy. Since experiments are all in simulation, it should be possible to analyze this.

- The experimental insights are not very interesting. The results show that BPPO is either very slightly better or slightly worse in all the environments of D4RL , so it is unclear why this algorithm should be preferred in practice over existing baselines. Also, D4RL seems to be a very saturated benchmark, particularly the Gym environments, so it is unclear whether a gain of reward by 1 point on Walker is a meaningful difference. Results for several algorithms on several environments are not presented in Table 1 - it should be simple to run them and perform comparisons, right?

- Offline RL depends on the quality of the offline dataset. There is no analysis showing how the proposed algorithm compares with baselines in different regimes of the optimality of offline data. I think this would be particularly important because the analysis depends on being able to recover the underlying behavior policy.

- In order to show that the proposed algorithm leads to non-trivial gains or is otherwise interesting empirically, I think there needs to be results in a practical setting where BPPO is able to "solve" a task while prior approaches fail to solve it. Alternatively, any other comparison that shows why BPPO should be used by practitioners - I am not convinced that simplicity alone can be the reason given prior works like TD3+BC are equally if not more simple and achieve nearly the same performance on the D4RL benchmark.

**Summary Of The Paper:**

This paper describes an empirical finding regarding conservatism of on-policy online RL, and uses this insight to propose an offline RL algorithm, Behavior PPO that monotonically improves the performance over behavior policy in the same manner as PPO during online training. The main contribution is that the proposed BPPO algorithm is simple to implement, achieves results comparable to prior works like CQL and IQL on D4RL benchmark, and has nice theoretical properties of monotonic improvement during training.

**Summary Of The Review:**

The paper provides an interesting theoretical insight and develop a practical simple algorithm, which is great, but the results are not at all convincing regarding why the proposed algorithm should be favored over prior works like CQL, IQL that also have neat theoretical insights (conservatism) and perform as well as the proposed algorithm on the D4RL benchmark, and prior works like TD3+BC that are equally simple to implement.

--- AFTER REBUTTAL RESPONSES ---

The authors have clarified some concerns regarding the experiments, and provided some missing results in the tables. As such I am updating my score to weak accept, but with low confidence, because the significance of the results are still unclear (minor improvements) and I am not convinced that the method is simpler to implement compared to the baselines (CQL, TD3+BC), and that the theoretical bounds on improvement are better than CQL.

---

> ### Author Response · Authors · 2022-11-17
> **Response to Reviewer QTZM (2/2)**
>
> **Q3: Since our analysis depends on being able to recover the underlying behavior policy, comparison in different regimes of optimality of offline data should be included.**
>
> **A3:** About the too strong assumption1, we have changed it into total variational divergence between estimated behavior policy and offline dataset (refer to proposition1). In the way, we no longer force the estimated behavior policy can recover the true behavior policy. We allow a certain gap and this will not affect the final theoretical findings. As for the comparison in different regimes of optimality benchmarks, the medium, medium-replay and medium-expert  in Gym are this kind of benchmarks.
>
> **Q4.1: Is BPPO able to solve some tasks while prior approaches fail to solve?**
>
> **A4.1:** BPPO superiorly solves tasks on Adroit, Kitchen and Antmaze compared to pervious work.
>
> **Q4.2: Why should BPPO be used by practitioners since TD3+BC is simple to implement?**
>
> **A4.2:** When comparing BPPO with TD3+BC, the common advantage is the simple implementation. But the performance of BPPO on D4RL is completely better than TD3+BC, especially on Adroit, Kitchen and Antmaze, that is why BPPO should be used by practitioners rather than TD3+BC.
>
> **Q5: (Clarity, Quality, Novelty And Reproducibility) The novelty of BPPO and the comparison between BPPO and Onestep RL.**
>
> **A5:** We have analyzed the relation of these two methods in subsection "What is the relation between BPPO and Onestep RL". Here I will give more comparisons in other aspects. Onestep RL intuitively regards the offline dataset $\mathcal{D}$ as the on-policy sample of estimated behavior policy $\hat\pi_{\beta}$ then calculate the behavior Q function to update $\hat\pi_{\beta}$. The theory of BPPO also starts from this but we theoretically measure this intuitive. Concretely, we thoroughly consider the gap between $\hat\pi_{\beta}$ and $\mathcal{D}$ (namely $\pi_{\beta}$). This gap will cause the two mismatches, $\rho_{\hat\pi_{\beta}}\neq\rho_{\mathcal{D}}$ and $A_{\hat\pi_{\beta}}\neq A_{\pi_{\beta}}$. We analyze these mismatches in theorem 2 and theorem 4 respectively which can also be considered as one theoretical explanation for Onestep RL.
>
> Besides, BPPO also consider how to improve the $\hat\pi_{\beta}$ many times (refer to theorem 3) and the comparison has shown the superiority of BPPO over its Onestep version (refer to sec 6.2 for experiment results). After detailed theoretical analysis, we derive BPPO, one algorithm that is simpler to implement and performs better on D4RL. In summary, although BPPO shares a similar intuitive with Onestep RL, BPPO is still novel enough and strongly contributes to the theoretical analysis and practical algorithm.
>
> **Q6: (Summary of the review) Why should BPPO be favored over prior works including CQL, IQL and TD3+BC?**
>
> **A6:** BPPO has solid theory, simple implementation and superior performance. Compared to CQL and IQL, we both have solid theoretical analysis, but BPPO is easier to implement and performs better on D4RL. Besides, our theory provides a new perspective, offline monotonic policy improvement, for solving offline RL. This theoretical insight is novel to understand and solve offline RL compared to CQL, IQL. Compared to TD3+BC, we both have simple implementation while BPPO enjoys solid and novel theoretical insight and superior performance. Totally, considering theory, implementation and performance at the same time, BPPO is the algorithm that should be favored.
>
> Thank you for your review again. We hope we have resolved your concerns and clarified the misunderstandings. We are always willing to answer any of your further concerns.

---

> ### Author Response · Authors · 2022-11-17
> **Response to Reviewer QTZM (1/2)**
>
> Thank the reviewer for the detailed comments. About assumption 1, we have realized the problem and assumption 1 has been replaced by another expression in latest submission (see **A1**, **A3** for more discussions). For the experimental problems and some possible misunderstandings, we have answered or clarified in **A2.1**, **A2.2**, **A2.3**, **A4.2**. Comparisons with previous work are presented in **A5** and **A6**.
>
> **Q1: Assumption 1 is too strong to achieve in practical settings.**
>
> **A1:** For the assumption1, we have replaced it with the distance between estimated behavior policy and offline dataset using total variational divergence (refer to proposition1) in the latest submission. This distance allows us to get rid of the artificial strong assumption that forces the closeness. At the same time, this modification does not affect the whole theory.
>
> **Q2.1: Why should BPPO be preferred in practice over existing baselines?**
>
> **A2.1**: We first briefly present the performance of BPPO on the benchmark D4RL (refer to table 1, 2 in latest submission) to clarify some misunderstanding. The performance on Gym is truly "slightly better or slightly worse" although the overall performance of BPPO is better. But the performance on Adroit and Kitchen is **completely superior** to existing algorithms (some missing results are presented in **A2.3**). Besides, we have added the results on Antmaze, BPPO achieves best results on five of six environments and only "Medium-play-v2" is not the best. So, the performance is relatively superior.
>
> | Environment       | TD3+BC  | Onestep | CQL     | IQL      | DT      | RvS-R   | RvS-G   | BC (Ours) | BPPO (Ours) |
> | ----------------- | ------- | ------- | ------- | -------- | ------- | ------- | ------- | --------- | ----------- |
> | Umaze-v2          | 78.6    | 64. 3   | 74.0    | 87.5     | 65.6    | 64.4    | 65.4    | 51.7      | **95.0**    |
> | Umaze-diverse-v2  | 71.4    | 60.7    | 84.0    | 62.2     | 51.2    | 70.1    | 60.9    | 48.3      | **91.7**    |
> | Medium-play-v2    | 10.6    | 0.3     | 61.2    | **71.2** | 1.0     | 4.5     | 58.1    | 16.7      | 51.7        |
> | Medium-diverse-v2 | 3.0     | 0.0     | 53.7    | **70.0** | 0.6     | 7.7     | 67.3    | 33.3      | **70.0**    |
> | Large-play-v2     | 0.2     | 0.0     | 15.8    | 39.6     | 0.0     | 3.5     | 32.4    | 48.3      | **86.7**    |
> | Large-diverse-v2  | 0.0     | 0.0     | 14.9    | 47.5     | 0.2     | 3.7     | 36.9    | 46.7      | **88.3**    |
> | *Total*           | *163.8* | *61.0*  | *303.6* | *378.0*  | *118.6* | *153.9* | *321.0* | *245.0*   | ***483.3*** |
>
> Thus, BPPO should be preferred in practice given the comparable performance on Gym, **completely superior** performance on Adroit and Kitchen, and relatively superior performance on Antmaze. In addition to superior experimental results, BPPO is simple to implement, which makes BPPO should be preferred.
>
> **Q2.2: Is the performance gain on a saturated benchmark is meaningful?**
>
> **A2.2:** We do consider the Gym environment is very "saturated" but other environments including Adroit, Kitchen and Antmaze are not. Therefore, the performance gain on Gym may not be meaningful while the higher performance on Adroit, Kitchen and Antmaze is valuable. BPPO does perform best on these not "saturated" environments.
>
> **Q2.3: Missing baselines should be presented.**
>
> **A2.3:** Since original paper does not present these results, our table is also blank. The missing baselines are presented as follows. For the result of Onestep RL, the results of "cloned" are from the original paper while the results of "human" and "kitchen" are from the method Onestep RL-esayBCQ. All the results of TD3+BC are conducted by the parameter $\alpha=4.0$. Unexpectedly, these results are not good (maybe this is the reason why original paper does not present the results).
>
> |                     | Onstep RL | TD3+BC | BPPO  |
> | ------------------- | --------- | ------ | ----- |
> | pen-human-v1        | 90.7      | 8.4    | 117.8 |
> | hammer-human-v1     | 0.2       | 2      | 14.9  |
> | door-human-v1       | -0.1      | 0.5    | 25.9  |
> | relocate-human-v1   | 2.1       | -0.3   | 4.8   |
> | pen-cloned-v1       | 60        | 41.5   | 110.8 |
> | hammer-cloned-v1    | 2         | 0.8    | 8.9   |
> | door-cloned-v1      | 0.4       | -0.4   | 6.2   |
> | relocate-cloned-v1  | -0.1      | -0.3   | 1.9   |
> | kitchen-complete-v0 | 2         | 0      | 91.5  |
> | kitchen-partial-v0  | 35.5      | 22.5   | 57    |
> | kitchen-mixed-v0    | 28        | 25     | 62.5  |

---

> ### Author Response · Authors · 2022-12-07
> **Response to Reviewer QTZM after rebuttal**
>
> Thanks to the reviewer for increasing the score after the rebuttal. We are very glad to be informed that some concern has been clarified. For the remained concerns, we make the following explanation:
>
> - **The significance of the results are still unclear (minor improvements):** We evaluate BPPO on a series of tasks, including the Gym locomotion, Adroit, Antmaze, and Kitchen from D4RL. The performance of BPPO on Gym locomotion is competitive in comparison with the SOTA methods. Besides, we stress that BPPO significantly outperforms the SOTA methods on Adroit, Antmaze, and Kitchen tasks that are considerably more difficult than the Gym tasks. In detail, the results of BPPO are increased by **61.7%**, **24.6%**, and **23.8%** on Adroit, Kitchen, and Antmaze tasks respectively. The superior performance demonstrates that BPPO can achieve stable improvements in diverse environments/tasks.
> - **Dose BPPO is simpler to implement compared to the baselines (CQL, TD3+BC)?** First, BPPO trains an estimated behavior $\hat\pi_\beta$, a Q-function $Q_{\pi_\beta}$ and a value function $V_{\pi_\beta}$ by standard behavior cloning, SARSA and regression over returns respectively. Then, BPPO uses the loss function of PPO to update the policy network, where the only difference is advantage replacement from $A_{\pi_k}$ to $Q_{\pi_\beta}-V_{\pi_\beta}$. Totally speaking, each part is easy to implement and properly combining these components leads to an effective offline algorithm. So, we believe BPPO is as simple as TD3+BC to implement and much simpler than CQL.
> - **Are the theoretical bounds on improvement better than CQL?** CQL proposes a bound to present the learned optimal policy that can achieve safe policy improvement over behavior policy. Given the different proof process, the bounds of CQL and BPPO are hard to compare directly.
>
> We hope that concerns can be eliminated. Thank you for your comments and score update again. We are always willing to discuss with you.

---

### Official Review · Reviewer_t4J4 · 2022-10-24

**Confidence:** 3
**Clarity, Quality, Novelty And Reproducibility:** good except that the code is not prov…
**Correctness:** 3
**Technical Novelty And Significance:** 3
**Empirical Novelty And Significance:** 3
**Recommendation:** 6

**Strength And Weaknesses:**

interesting finding, good theoretical analysis, and sound experiments.

**Summary Of The Paper:**

The paper finds that some online on-policy algorithms are naturally able to solve offline RL by theoretical analysis. PPO method is extended to the offline setting by an additional model of advantage. Experiments show that the proposed method achieves good performance without expensive online interaction.

**Summary Of The Review:**

The paper is well-written and meaningful. The finding is interesting.  I would like to ask some questions about the method:
1. Since a model is introduced to estimate the advantage, are there any experiments to show the accuracy of the model?
2. For Eq. 13, the loss of \pi_k is only related to the behavior policy and \pi_k itself. In other words, \pi_k is not improved based on \pi_k-1. Then, why does the algorithm need K steps to get the best policy?
3. For Algorithm 1, what is the meaning of \pi <= \pi_k in line 7? is it measured by the return of the policy?

---

> ### Author Response · Authors · 2022-11-17
> **Response to Reviewer t4J4**
>
> Thank reviewer for the approval. All the questions have been detailed answered as follows.
>
> **Q1: Since a model is introduced to estimate the advantage, are there any experiments to show the accuracy of the model?**
>
> **A1:** We replace the Generalized Advantage Estimation (GAE) with $A_{\pi_{\beta}}\left(s,a\right) = Q_{\pi_{\beta}}\left(s,a\right) - V_{\pi_{\beta}}\left(s\right)$ because GAE is unable to generalize to unseen state-action pairs that are outside the dataset. Therefore, we are unable to compare the accuracy because most advantages of state-action pairs are unable to predict by GAE. But the superior performance can indirectly prove the accuracy.
>
> **Q2.1: For Equation (13), the loss of $\pi_k$ is only related to the behavior policy and $\pi_k$ itself. In other words, $\pi_k$ is not improved based on $\pi_{k-1}$.**
>
> **A2.1:** In Equation (13) of previous submission (refer to equation (17) in latest submission), we aim to improve the policy $\pi_k$ **rather than** improve $\pi_{k-1}$ to get a better $\pi_k$.
>
> **Q2.2: Then, why does the algorithm need $K$ steps to get the best policy?**
>
> **A2.2:** We apologize for the unclear description of the workflow in Algorithm 1. We have rewritten the pseudo-code step-by-step in the latest submission. For BPPO, the old policy $\pi_k$ is replaced by the target policy $\pi$ when the performance of the target policy improves. Therefore, $K$ denotes the number of policy replacement. After improving over behavior policy $\hat\pi_{\beta}$, the learned policy can not be guaranteed to optimal. So, we improve $K$ steps, trying to make $\pi_k$ extract all the useful information from offline dataset. This question also corresponds to the superiority of BPPO over its Onestep version. More discussion can be found in "What is the relation between BPPO and Onestep RL?" and section 6.2.
>
> **Q3: For Algorithm 1, what is the meaning of $\pi \leq \pi_k$  in line 7? is it measured by the return of the policy?**
>
> **A3:** Thanks for the reviewer’s careful check. We apologize for the unclear description of the workflow in Algorithm 1. We have replaced it by $J(\pi)$ to denote the measurement of the policy performance. Please check the rewritten Algorithm 1 for more details in the latest submission.
>
> Thank you for your review again. We hope we have resolved your concerns. We are always willing to answer any of your further concerns.

---

> ### Author Response · Authors · 2022-12-10
> **Response to Reviewer t4J4 during rebuttal**
>
> Since the Reviewer-Author discussion phase is coming to an end, we want to know whether reviewer has further suggestions for our paper. If any, please let us now. We are always willing to discuss with you.

---

### Official Review · Reviewer_7X1S · 2022-10-25

**Confidence:** 4
**Correctness:** 3
**Technical Novelty And Significance:** 3
**Empirical Novelty And Significance:** 3
**Recommendation:** 5

**Clarity, Quality, Novelty And Reproducibility:**

$\textbf{Clarity: }$

The paper is well written and the presentation of the proposed algorithm is very clear.

$\textbf{Novelty: }$

To my knowledge, most part of the theoretical results and algorithm designs are not new to me, since I view them as a concrete form in offline RL which can be easily derived from GePPO (i.e., when the offline buffer is considered as the off-policy data). Using the advantage estimates of behavior policy to bypass the need of estimating the advantage of current policies is novel. However, I have a major concern on the soundness of this replacement.

$\textbf{Quality: }$

Assumption 1 is difficult to achieve in practice to my knowledge. Moreover, Theorem 2 may have a small issue in derivation due to a mismatch between the required condition and Assumption 1. The proposed algorithm is estabilished on two approximation choices which can be very loose in the policy improvement bound. For the experiments,


$\textbf{Reproductibility: }$

The proposed algorithm is clear and it seems to be easy to implement. However, the source codes are not provided.


**Strength And Weaknesses:**

$\textbf{Strengths:}$
+ I appreciate the authors' efforts in study the monotonic policy improvement in offline RL. To my knowledge, it is novel.
+ The paper is well written and the presentation of the proposed algorithm is very clear.
+ The experiments are conducted from multiple aspects.

&nbsp;

$\textbf{Weaknesses:}$

Although I think the authors effort in studying monotonic policy improvement in offline RL is novel, I view it as a specific case of Generalized Off-policy Monotonic Policy Improvement.

Two representative prior works are:
- [1] James Queeney, Yannis Paschalidis, Christos G. Cassandras. Generalized Proximal Policy Optimization with Sample Reuse. NeurIPS 2021
- [2] Wenjia Meng, Qian Zheng, Yue Shi, Gang Pan. An Off-Policy Trust Region Policy Optimization Method With Monotonic Improvement Guarantee for Deep Reinforcement Learning. IEEE Trans. Neural Networks Learn. Syst. 33(5): 2223-2235 (2022)

&nbsp;


In my view, Theorem 1 and 2 presented in this paper are specific forms of the Generalized Policy Improvement Lower Bound as presented by Theorem 1 in GePPO [1] (similar ones can also be found in [2]). Concretely, if we use the offline buffer for the off-policy state distribution considered in Theorem 1 in GePPO, we can derive very similar form to Theorem 1 and 2 (maybe only differ at the use of $\xi$-coupled policy for expression).

Therefore, the theoretical results are not new to me. I think it is very necessary to include the two works mentioned in the related work and maybe the methodology of this paper, as they are not mentioned in the “Monotonic Policy Improvement” paragraph.


&nbsp;


For using the advantage estimates of the behavior policy to bypass the advantage estimation of iterative $\pi_k$ is new to me. However, I worry about the soundness since the approximation makes the policy improvement lower bound even looser.

The significant performance difference between $\omega=0.9$ and $\omega=0.5$ of the asymmetric coefficient in Figure 4 may support my concern on the soundness of using $\bar A_{\hat{\pi}_{\beta}}$.

&nbsp;


Theorem 2 and 3 are based on Assumption 1. It seems that there is mismatch between the state-action support considered in Assumption 1 and in Lemma 1, 2, i.e., $\xi$-coupled or $\alpha$-coupled policy for the finite states in the offline buffer v.s. for all states. Is this mismatch a small issue in the derivation?

&nbsp;


Besides, what does $\pi \le \pi_k$ condition at Line 7 in Algorithm 1 mean?


&nbsp;

For the expeirments, in addition to the two hyperparameters investigated in Section 6.3, I think readers are also interested in how the iterative step number $K$ in Algorithm 1 and the decay coefficient/decay schedule are selected and designed, and how different choices of them influence the performance.


I appreciate the empirical comparison in Section 6.2. Since Onestep BPPO is the variant let $K=0$ in Algorithm 1, one question here is, what if we update policy $\pi$ by maximizing $\hat{L}_{\beta}(\pi)$, i.e., Equation 10, for the same number of BPPO? I think some additional results for this will strengthen the comparison in Section 6.2.


To make the performance comparison in Section 6.1 more complete, I recommend the authors to consider EDAC [3], LAPO [4] and RORL [5] as SOTA baselines later.


- [3] Gaon An, Seungyong Moon, Jang-Hyun Kim, Hyun Oh Song. Uncertainty-Based Offline Reinforcement Learning with Diversified Q-Ensemble. NeurIPS 2021
- [4] Xi Chen, Ali Ghadirzadeh, Tianhe Yu, Yuan Gao, Jianhao Wang, Wenzhe Li, Bin Liang, Chelsea Finn, Chongjie Zhang. Latent-Variable Advantage-Weighted Policy Optimization for Offline RL. arXiv: 2203.08949 (2022)
- [5] Rui Yang, Chenjia Bai, Xiaoteng Ma, Zhaoran Wang, Chongjie Zhang, Lei Han:
RORL: Robust Offline Reinforcement Learning via Conservative Smoothing. arXiv:2206.02829 (2022)





**Summary Of The Paper:**

This paper studies monotonic policy improvement in offline RL, aiming at demonstrating the effectiveness of online monotonic policy improvement algorithm in solving offline RL. Following the vein of TRPO and PPO, this work proposes Behavior Proximal Policy Optimization (BPPO) by adjusting the policy improvement lower bound based on the states in offline buffer and the advantage estimates of behavior policy. The proposed algorithm is evaluated on the D4RL benchmark, consisting of the performance comparsion with several onestep offline RL and iterative offline RL algorithms and the analysis of two hyperparameters.

**Summary Of The Review:**

According to my detailed review above, I think this paper is below the acceptance threshold (actually I would give a 4 if there was).

This is mainly due to the overlap between the proposed theories in this work and GePPO (and other off-policy TRPO/PPO advances), my concern on the soundness of the advantage approximation and the lack of important hyparameter analysis in the experiments.

---

> ### Author Response · Authors · 2022-11-17
> **Response to Reviewer 7X1S (2/2)**
>
> **Q5.2: The decay coefficient/decay schedule are selected and designed, and how different choices of them influence the performance.**
>
> **A5.2:** For the decay coefficient, we have conducted a couple of experiments to explore how it contributes to the performance of the BPPO and Onestep BPPO in Section 6.3 and Appendix H. In Appendix G, we show why the decay coefficient is important to maintain the BPPO in a high-quality performance. *Please see the latest submission for more details.*
>
> **Q7: To make the performance comparison in Section 6.1 more complete, I recommend the authors to consider EDAC [3], LAPO [4] and RORL [3] as SOTA baselines later.**
>
> **A7:** We have added the EDAC [3], LAPO [4] and RORL [5] as comparison baseline in the latest submission in Appendix I. The performance of BPPO is slightly worse than the EDAC [3], and RORL [5] on gym environments while BPPO significantly outperforms all methods on the Adroit, Kitchen, and Antmaze environments.
>
> | Environment                  | [3]     | [5]     | BPPO    |
> | ---------------------------- | ------- | ------- | ------- |
> | halfcheetah-medium-v2        | 65.9    | 66.8    | 44.0    |
> | hopper-medium-v2             | 101.6   | 104.8   | 93.9    |
> | walker2d-medium-v2           | 92.5    | 102.4   | 83.6    |
> | halfcheetah-medium-replay-v2 | 61.3    | 61.9    | 41.0    |
> | hopper-medium-replay-v2      | 101.0   | 102.8   | 92.5    |
> | walker2d-medium-replay-v2    | 87.1    | 90.4    | 77.6    |
> | halfcheetah-medium-expert-v2 | 106.3   | 107.8   | 92.5    |
> | hopper-medium-expert-v2      | 110.7   | 112.7   | 112.8   |
> | walker2d-medium-expert-v2    | 114.7   | 121.2   | 113.1   |
> | *Gym locomotion-v2 total*    | *841.1* | *870.8* | *751.0* |
> | pen-human-v1                 | 52.1    | 33.7    | 117.8   |
> | hammer-human-v1              | 0.8     | 2.3     | 14.9    |
> | door-human-v1                | 10.7    | 3.8     | 25.9    |
> | relocate-human-v1            | 0.1     | 0.0     | 4.8     |
> | pen-cloned-v1                | 68.2    | 35.7    | 110.8   |
> | hammer-cloned-v1             | 0.3     | 1.7     | 8.9     |
> | door-cloned-v1               | 9.6     | -0.1    | 6.2     |
> | relocate-cloned-v1           | 0.0     | 0.0     | 1.9     |
> | *adroit-v1 total*            | *141.8* | *77.1*  | *291.4* |
>
> | Environment | [4]     | Ours    |
> | ------------------- | ------- | ------- |
> | kitchen-complete-v0 | 53.2    | 91.5    |
> | kitchen-partial-v0  | 53.7    | 57.0    |
> | kitchen-mixed-v0    | 62.4    | 62.5    |
> | *kitchen-v0 total*  | *169.3* | *211.0* |
>
> | Environment | [5]     | Ours    |
> | ------------------ | ------- | ------- |
> | Umaze-v2           | 96.7    | 95.0    |
> | Umaze-diverse-v2   | 90.7    | 91.7    |
> | Medium-play-v2     | 76.3    | 51.7    |
> | Medium-diverse-v2  | 69.3    | 70.0    |
> | Large-play-v2      | 16.3    | 86.7    |
> | Large-diverse-v2   | 41.0    | 88.3    |
> | *Antmaze-v2 total* | *390.3* | *483.3* |
>
> [1] James Queeney, Yannis Paschalidis, Christos G. Cassandras. Generalized Proximal Policy Optimization with Sample Reuse. NeurIPS 2021
>
> [2] Wenjia Meng, Qian Zheng, Yue Shi, Gang Pan. An Off-Policy Trust Region Policy Optimization Method With Monotonic Improvement Guarantee for Deep Reinforcement Learning. IEEE Trans. Neural Networks Learn. Syst. 33(5): 2223-2235 (2022)
>
> [3] Gaon An, Seungyong Moon, Jang-Hyun Kim, Hyun Oh Song. Uncertainty-Based Offline Reinforcement Learning with Diversified Q-Ensemble. NeurIPS 2021
>
> [4] Xi Chen, Ali Ghadirzadeh, Tianhe Yu, Yuan Gao, Jianhao Wang, Wenzhe Li, Bin Liang, Chelsea Finn, Chongjie Zhang. Latent-Variable Advantage-Weighted Policy Optimization for Offline RL. arXiv: 2203.08949 (2022)
>
> [5] Rui Yang, Chenjia Bai, Xiaoteng Ma, Zhaoran Wang, Chongjie Zhang, Lei Han: RORL: Robust Offline Reinforcement Learning via Conservative Smoothing. arXiv:2206.02829 (2022)
>
> [6] Achiam J, Held D, Tamar A, et al. Constrained policy optimization[C]//International conference on machine learning. PMLR, 2017: 22-31.
>
> Thank you for your review again. We hope we have resolved your concerns. We are always willing to answer any of your further concerns.

---

> ### Author Response · Authors · 2022-11-17
> **Response to Reviewer 7X1S (1/2)**
>
> Thank the reviewer for the high praise and detailed suggestions from the perspective of monotonic policy improvement. The proposed references [1] [2] help us improve the quality of our proof. About the review's most concerned problem, advantage replacement, the discussion is presented in **A2**. We hope we can resolve your concerns.
>
> **Q1: Missing related work about monotonic policy improvement.**
>
> **A1:** Let me appreciate again for the reviewer's guidance on monotonic policy improvement. We have added these two papers [1] [2] into the related work. Furthermore, some proof techniques from [1] [2] help our bound tighter (refer to theorem 2, 3, 4). Another thing we would like to clarify is because offline RL is an extreme off-policy case, the overlap between our theory and GePPO, more strictly speaking, is an avoidable fact rather than something like weakness. Besides the overlap part, the advantage replacement (refer to theorem4) is unique in offline setting.
>
> **Q2: The soundness about the advantage replacement.**
>
> **A2:** In online setting, advantage $A_{\pi_k}$ can be calculated by GAE using the trajectories collected by policy $\pi_k$. By comparison, offline RL has to consider how to approximate advantage $A_{\pi_k}$ using the offline dataset $\mathcal{D}$ collected by the behavior policy $\pi_{\beta}$, which is the unique challenge of offline RL. We have two approaches to solving this problem, theoretically or empirically.
>
> The theoretical approach is the advantage replacement (refer to theorem4) utilized by BPPO, which replaces the $A_{\pi_k}$ with $A_{\pi_{\beta}}$. This does make the bound looser, but we are able to accurately estimate the $A_{\pi_{\beta}}$, that is, all the possible error has been measured by theorem 4. Another empirical one is calculating $A_{\pi_k}$ using $\mathcal{D}$ by off-policy method, which may suffer from overestimation problem. Thus, additional method has to be introduced to solve overestimation.
>
> From the perspective of monotonic policy improvement, the empirical approach should be preferred due to the tighter bound. But from the perspective of offline RL, the empirical one is potentially uncontrollable and difficult to be measured while the theoretical approach has considered the potential risk which makes empirical advantage easy to implement. Furthermore, one of our motivations is to show some online on-policy method is able to solve offline RL without extra constraint or regularization introduced. In summary, the advantage replacement (refer to theorem 4) is not the best choice from the viewpoint of theory but comprehensively considering theoretical results, empirical implementation and motivation, the advantage replacement is better.
>
> **Q3: Does the mismatch between the state distribution in assumption 1 and in lemma 1, 2 is a small issue?**
>
> **A3:** In the latest submission version, we remove the assumption1 and measure the distance between estimated behavior policy and offline dataset using total variational divergence (refer to proposition 1). The state mismatch in previous version is more like a clerical error since our method only accesses the state inside the offline dataset and "all the states" is not required. And in latest submission, the $\alpha$-coupled technique is replaced by lemma 2 in Appendix C from CPO [6].
>
> **Q4: what does condition $\pi \leq \pi_k$ at Line 7 in Algorithm 1 mean?**
>
> **A4:** Thanks for the reviewer’s careful check. We apologize for the unclear description of the workflow in Algorithm 1. We have replaced it with $J(\pi)$ to denote the measurement of the policy performance. Please check the rewritten Algorithm 1 for more details in the latest submission.
>
> **Q5.1 & Q6: (Q5.1) For the experiments, in addition to the two hyperparameters investigated in Section 6.3, I think readers are also interested in how the iterative step number K in Algorithm 1. (Q6) Since Onestep BPPO is the variant let in Algorithm 1, one question here is, what if we update policy $\pi$ by maximizing $L_{\hat{\pi}_\beta}$, i.e., Equation (10), for the same number of BPPO?**
>
> **A5.1 & A6:** Thanks for the insightful suggestion. We apologize for the unclear description of the workflow in Algorithm 1. We have rewritten the pseudo-code step-by-step in the latest submission. In the comparison experiment, both the BPPO and Onestep BPPO are trained by $I$ gradient steps. For BPPO, the old policy $\pi_k$ is replaced by the target policy $\pi$ when the performance of the target policy improves. Therefore, $K$ denotes the number of policy replacement. For Onestep BPPO, we train the policy by **the same gradient steps as the BPPO**, but do not update the behavior policy $\hat\pi_\beta$, i.e., removing steps 8-10 in Algorithm 1.

---

> ### Author Response · Authors · 2022-12-10
> **Response to Reviewer 7X1S during rebuttal**
>
> Here is a kind reminder that the Reviewer-Author discussion phase is coming to an end. We are very grateful for the valuable suggestions and guidance about monotonic policy improvement theory. At the same time, we also expect your further suggestions on our response and modification. Here are the key questions:
>
> - **Theory Part:** The overlap between offline monotonic policy improvement and off-policy case is reasonable and unavoidable because offline is a specific case of off-policy. The biggest difference is the estimation of $A_{\pi_k}$. For online setting, $A_{\pi_k}$ can be easily estimated through interaction with environment. But in offline situation, how to accurately estimate $A_{\pi_k}$ with $\mathcal{D}$ is **challenging** due to the potential overestimation. Therefore, we choose to replace $A_{\pi_k}$ with $A_{\pi_{\beta}}$. Totally, advantage replacement dose loose the bound which shouldn't be preferred from the perspective of theory, but when consider theory and practical implements simultaneously, we believe advantage replacement is better (more discussion please refer to (**Q1**&**A1** and **Q2**&**A2**).
> - **Experiment Part:** Missing ablation study has been supplemented in section 6.3, Appendix H and Appendix G (refer to  **Q5.2**&**A5.2**). Recommended comparison can be found in Appendix I (also refer to **Q7**&**A7**). We also supplement experiments on Antmaze in section 6.1. The performance of BPPO on Gym is competitive while BPPO significantly outperforms the SOTA methods on Adroit (*61.7%*), Antmaze (*24.6%*), and Kitchen (*23.8%*) tasks that are considerably more difficult than the Gym.
>
> We hope we can resolve the concern about the advantage replacement. We hope to hear your further comments and discuss with you.

---

### Official Review · Reviewer_qeAt · 2022-10-25

**Confidence:** 3
**Correctness:** 2
**Technical Novelty And Significance:** 3
**Empirical Novelty And Significance:** 3
**Recommendation:** 3

**Clarity, Quality, Novelty And Reproducibility:**

The paper is mostly clear, apart from some sections that need to be further polished (see weaknesses). The theory seems of high quality, but the practical approximations make the proposed algorithm less sound. Further experiments are needed to improve the quality of the paper. To my knowledge, the results are novel. The authors specify the hyperparameters used in their implementation.

**Strength And Weaknesses:**

Strengths:
- The theoretical results seem correct and not trivial. The authors state the assumptions well before each theorem.
- The problem is significant. In particular, recent work showed that applying on-policy algorithms to offline RL problems can lead to surprising results. This paper goes one step further in such a direction.
- The experimental results show higher performance with respect to the baselines

Weaknesses:
- Some parts of the paper seem written in a hurry and are unclear. For example, the first 5 lines after Eq. 11 are confusing. Other parts of the paper seem too colloquial. For instance "simply augmenting TD3 (...) with behavior cloning (...) reminds us to revisit", or "the most tough thing is the existence of A".
- There are claims that are not verified through experiments. In the abstract and in the introduction, it is stressed that existing off-policy actor-critic methods overestimate out-of-distribution actions. This motivates the use of a proximal objective. However, the paper is lacking an experiment showing that the proposed method does not suffer from this issue.
- I have a few concerns regarding the soundness of the method. The sequential optimization of bound (8) would lead to policy improvement. However, the approximation in Eq 13 has no guarantees to improve upon the behavioral policy. The only guarantees are given by Theorem 4, which does not involve the true objective. Can the authors clarify this?
- The comparison with Onestep BPPO, which consists of Behavioral Cloning + PPO, is not convincing. Did the authors take the tuned hyperparameters they found for their method and applied them to the Onestep method?
- The paper is lacking an important ablation on the clip coefficient decay and the asymmetric coefficient for the advantage: How does the method perform without such tricks? How does it perform with respect to Onestep PPO, when both methods are without tricks?
- I kind of disagree with the authors in the main claim of the paper, which is that they discover that an online, on-policy method can solve offline RL problems. What the paper proposed, is instead an off-policy version of PPO that works for offline RL. Please note that PPO can already be considered an off-policy algorithm, since at each iteration, after the first policy gradient step, it is learning about a policy that was not used to collect the data. The clipped importance weight helps keep the optimization to be near-on-policy. BPPO further relaxes this constraint, letting the learned policy be able to be far from the behavioral policy.

Other questions:
- The experiments involve continuous action spaces. How does this affect Assumption 1? I expect the value of $\xi$ to be 1.
- What is the value of the bounds proposed during training? What is the magnitude of each component in the bound during the experiments?
- At each iteration, the importance weight between the current policy and the previous one is clipped. Can the authors quantify instead the importance weight between the final learned policy and the behavioral policy? Can this value be arbitrarily large?

**Summary Of The Paper:**

This paper addresses the problem of learning in the offline RL scenario. While existing approaches need to regularize objectives in order to learn a policy in proximity to the one used to collect data, the authors explore the use of online on-policy algorithms in order to solve such tasks. Their focus is on the policy improvement theorem, from which notable algorithms like TRPO or PPO can be derived. They first extend such a theorem to the offline setting, deriving a lower bound on the true performance that depends on additional terms related to the offline dataset. Then they extend such theorem to consider an improvement over an arbitrary learned policy. Based on these results, they derive an iterative algorithm that optimizes the PPO objective, but with a change in Importance Sampling ratios, where the denominator is constantly replaced by the current learned policy. To prevent this optimization process to diverge, they introduce a clipping decay mechanism.

Experiments show that this approach is effective in solving offline tasks. Moreover, an ablation on different hyperparameters demonstrates the robustness of the method.

**Summary Of The Review:**

This paper introduces an offline version of PPO that can be used to solve offline tasks. It is based on an offline version of the policy improvement theorem. Although the experiments show improvement, there are a few concerns regarding the soundness of the proposed approach and the experimental choice before acceptance.

---

> ### Author Response · Authors · 2022-11-17
> **Response to Reviewer qeAt (2/2)**
>
> **Q6: Disagreement about the main claim: "an online on-policy method can solve offline RL problems".**
>
> **A6:** We first clarify our understanding of PPO. At each policy improvement iteration of PPO, the current policy must collect trajectories then use these trajectories to improve the policy. From the perspective of data collection in each iteration, PPO is an on-policy method. However, we also note that PPO often performs several updates and thus importance sampling is needed. From this perspective, PPO is an off-policy style method. Considering these two points simultaneously, PPO is a *near-on-policy* method.
>
> Our main claim aims to emphasize two points. One is "online", which emphasizes our method makes no modification to the loss function of PPO. Another is "on-policy", which emphasizes that most previous work focus on how to make off-policy method work again in offline setting while we find some on-policy algorithm naturally works due to the inherent conservatism (refer to equation (17), (20)). But there must exist some mismatch between PPO and offline setting. We solved this mismatch both theoretically and empirically then proposed BPPO. Another misunderstanding we want to clarify is "the learned policy is able to be far from the behavioral policy". Actually, the learned policy still close to the behavioral policy and more detailed discussion is presented in **A9**.
>
> **Q7: Does assumption1 is reasonable in practice?**
>
> **A7:** We have removed this strong assumption which forces the closeness and introduce another distance as alternative (refer to proposition 1). This distance is determined by offline dataset only and we no longer require the closeness. Besides, based on this distance, we can also derive the same theoretical results.
>
> **Q8: What is value of the bounds during training? What is the magnitude of each component in the bound during the experiments?**
>
> **A8:** The accurate bounds are impossible to calculate due to the existence of the maximum total variational distance. So the value of each component is unable to present.
>
> **Q9: Can the authors quantify the importance weight between the final learned policy and the behavioral policy? Can this value be arbitrarily large?**
>
> **A9:** To answer this question, we consider exploring whether the importance weight between the improved policy $\pi_k$ and the behavior policy $\pi_{\beta}$ will be arbitrarily large. To this end, we quantify this importance weight in the training phase in Figure 6 in Appendix G in the latest submission. In Figure 6, we observe that the ratio of the BPPO with decay always stays in the clipped region (the region surrounded by the dotted yellow and red line). Therefore, the importance weight between the final learned policy and the behavioral policy is **not arbitrarily large**. However, the BPPO without decay is beyond the region in Figure 6(a) and 6(b). It demonstrates the improved policy without decay is farther away from the behavior policy than the case of BPPO with decay. It may cause unstable performance and even crashing, as shown in Figure 4 and 8 when $\sigma=1.00$ (i.e., without decay). *Please see the latest submission for more details.*
>
> [1] Fujimoto, S., & Gu, S. S. (2021). A minimalist approach to offline reinforcement learning. *Advances in neural information processing systems*, *34*, 20132-20145.
>
> Thank you for your review again. We hope we have resolved your concerns. We are always willing to answer any of your further concerns.

---

> ### Author Response · Authors · 2022-11-17
> **Response to Reviewer qeAt (1/2)**
>
> Really appreciate for your comprehensive comments. All the questions have been carefully answered.
>
> **Q1: Some parts of this paper seem unclear, and more interpretation is needed.**
>
> **A1:** The first 5 lines after equation (11) (in previous version) are explaining why we adopt $A=Q-V$ to calculate advantage rather than GAE. In the new submission version, this part refers to the "Advantage Approximation" in section 4 after equation (10). All the colloquial expression has been polished.
>
> **Q2.1: Existing off-policy actor-critic methods overestimate out-of-distribution actions. This motivates the use of a proximal objective.**
>
> **A2.1:** Existing off-policy actor-critic methods suffer from overestimation is **not** why we use a proximal objective. The proximal objective is strictly derived from offline monotonic policy improvement. There is no direct causal relationship between these two things.
>
> **Q2.2: The paper lacks experiment to show that the proposed method does not suffer from the overestimation of out-of-distribution actions.**
>
> **A2.2:** Existing off-policy actor-critic is based on policy iteration. In policy evaluation, the Q-function may poorly estimate the value of out-of-distribution state-action pairs and this affects policy improvement, where the policy takes the action with highly estimated Q-values [1]. BPPO is **not** based on the policy iteration and BPPO first learns an on-policy advantage $A=Q-V$ then updates the estimated behavior policy for $K$ times. BPPO constrains the learned policy close to the behavior policy so BPPO will not access to the out-of-distribution actions. Therefore, the overestimation of out-of-distribution is impossible to occur. For experiments, our superior performance on D4RL indirectly proves the overestimation does not happen.
>
> **Q3: I have a few concerns regarding the soundness of the method. The sequential optimization of bound (8) would lead to policy improvement. However, the approximation in Eq 13 has no guarantees to improve upon the behavioral policy. The only guarantees are given by Theorem 4, which does not involve the true objective. Can the authors clarify this?**
>
> **A3:** Very sorry to confuse reviewer and we have made it clear in latest submission. Combing the result of theorem 3 (namely the bound (8) in pervious version) and theorem 4, we derive one corollary, which involves the true objective. Furthermore, **conclusion3** directly give the guarantee to improve the policy $\pi_k$ (when $k=0$, $\pi_k$ is the behavior policy $\hat\pi_{\beta}$ ). Based on the **conclusion3**, the loss function (17) is derived (corresponding to the equation (13) in pervious version).
>
> **Q4: Does the authors take the tuned hyperparameters they found for their method and apply them to the Onestep version?**
>
> **A4:** We apologize for not stating the experiment set in the previous submission. The hyperparameters of both the Onestep BPPO and BPPO are tuned through the grid search, then we exhibit their learning curves with the best performance in Figure 2. Besides, we have added the relative ablation analysis of Onestep BPPO on the hyperparameters in Appendix H of the latest submission. *Please see the latest submission for more details.*
>
> **Q5: The paper is lacking an important ablation on the clip coefficient decay and the asymmetric coefficient for the advantage: How does the method perform without such tricks? How does it perform with respect to Onestep PPO, when both methods are without tricks?**
>
> **A5:** We have added the ablation studies on the clip coefficient and asymmetric coefficient for Onestep BPPO in Appendix H of the latest submission. **Note that**, the clip coefficient decay rate $\sigma=1.0$ and the asymmetric coefficient $\omega=0.5$ denote those tricks are removed, respectively. *Please see the latest submission for more details.*

---

> ### Author Response · Authors · 2022-12-10
> **Response to Reviewer qeAt during rebuttal**
>
> Here is a kind reminder that the Reviewer-Author discussion phase is coming to an end. We'd like to know whether our response has resolved your concerns and whether our modification has improved the paper. For the most concerned problems and corresponding modification, we briefly summarize as follow:
>
> - (**Q7**&**A7**) For assumption 1 which is difficult to achieve in practice, we have change it into one distance which is only related to the offline dataset (please refer to proposition 1). The change will not affect our final theoretical results.
> - (**Q2.2**&**A2.2**) BPPO forces the learned policy close to the behavior policy indirectly, so overestimation of out-of-distribution actions can be eliminated (please refer to page 6 "why BPPO can solve offline RL?" for more detail).
> - (**Q9**&**A9**) The importance weight between the final learned policy and the behavioral policy is **not arbitrarily large** (please refer to Appendix G).
> - Supplemented experiments can be find in (**Q4**&**A4**) and (**Q5**&**A5**).
>
> Your further comments and suggestions are very important to us, and we hope to hear you soon.

---

### Author Response · Authors · 2022-11-17
**Modification about the New Submission**

Thanks for reviewer's detailed suggestions and comments. Given these, we have updated our submission and the major modifications are summarized as follows:

- Since four reviewers (of five) expressed the concern about the assumption1 which is too strict to accomplish in practice, we removed this assumption. As alternative, we use the total variational divergence to measure the distance between estimated behavior policy and offline dataset (refer to proposition 1). This distance is only determined by the offline dataset rather than a strong artificial assumption. Changing assumption 1 into this distance does not affect the theoretical results.
- Besides the assumption 1 has been changed, we also update our proof process. Concretely, the $\alpha$-coupled assumption (previously presented in the appendix) has been replaced by one inequality from CPO [1]. New proof techniques result in tighter bounds (refer to theorem 2, 3, 4). One corollary is added after theorem 4, which more clearly describes that how to monotonically improve the policy using offline dataset.
- When deriving the loss function based on the theoretical results, we have added more detailed process. In this way, why BPPO can solve offline RL and how to constrain the closeness between learned policy and behavior policy are clearer (refer to equation (17), (20)).
- Supplemented experiments include: **a)** In Section 6.1, we evaluate the performance of the BPPO on Antmaze tasks with sparse reward settings and demonstrate the superiority of the BPPO compared with previous offline algorithms. **b)** In Section 6.3, we demonstrate how the clip ratio decay affects the performance of the BPPO and Onestep BPPO. **c)** In Appendix G, we quantify the importance weight in the training phase to explore whether the importance weight between the learned policy and the behavior policy $\hat\pi_\beta$ will be arbitrarily large. Then we further analyze why the clip ratio is important for BPPO. **d)** In Appendix H, we analyze how the clip ratio $\epsilon$, the clip ratio decay $\sigma$, and the asymmetric coefficient $\omega$ affect the performance of the Onestep BPPO. **e)** In Appendix I, we have added more SOTA methods as the comparison baselines to further evaluate the superiority of the BPPO.
- We promise that we will make our code public after the final decision.

[1] Achiam J, Held D, Tamar A, et al. Constrained policy optimization[C]//International conference on machine learning. PMLR, 2017: 22-31.

---

### Author Response · Authors · 2022-11-28
**Overall Response to All Reviewers**

We would like to sincerely appreciate the reviewers for the insightful suggestions. Regardless of the positive or negative rating, all the comments have greatly helped us to further polish the paper. Here we summarize and respond to the issues that the reviewers mainly focused on.

For the theoretical analysis, the strong assumption1 and the replacement of $A_{\pi_k}$  with $A_{\pi_\beta}$  are two most concerned issues:

- Four reviewers (of five) think the assumption1 is too strong to accomplish. So, we replace it with the total variational divergence between estimated behavior policy and offline dataset (refer to proposition 1). This distance is only determined by the offline data rather than a strong artificial assumption. This modification does not affect the final theoretical results.
- We admit that replacing $A_{\pi_k}$  with $A_{\pi_\beta}$  does loose the bound while we want to emphasize the necessity of this replacement. If we calculate the $A_{\pi_k}$  by some off-policy method, the potential overestimation should be carefully considered. But this is more empirically uncontrollable compared to advantage replacement. In other words, we sacrifice some theoretical performance in exchange of the overall performance of our method.

For the experimental evaluations, we have added the necessary ablation study of clip ratio decay $\epsilon$ in section 6.3. The performance on tasks with sparse reward (Antmaze) has been presented on section 6.1. Other supplemented experiments are listed in "Modification about the new paper version".

For the novelty or contribution of our paper, we provide the following statement. We first propose offline monotonic policy improvement to formulate the offline reinforcement learning (theoretical contribution), then we derive one simple and effective algorithm called Behavior Proximal Policy Optimization (empirical contribution). Compared to Onestep RL, our novelty is still enough. Part of our theory can theoretically explain the intuition of it (refer to theorem 1) and our implementation is quite different especially "BPPO is neither Onestep nor multi-step" (refer to page 6 subsection "What is the relation between BPPO and Onestep RL?").

We hope this brief summary has highlighted the most concerned questions and our corresponding solutions. We are always willing to answer any of your further concerns and adopt your insightful suggestions to improve our paper.

---

### Decision · Program_Chairs · 2023-01-20

**Decision:**

Accept: poster

**Justification For Why Not Higher Score:**

The paper does not show performance that is much better than different approaches. It is more of interest to the community because it shows that online on-policy algorithms seem to be naturally able to solve offline RL problems.

**Justification For Why Not Lower Score:**

The results and findings are likely of interest to the community.

**Metareview: Summary, Strengths And Weaknesses:**

This paper is based on the observation that online on-policy algorithms are naturally able to solve offline RL problems. It provides a theoretical explanation and extensive experimental results. While reviewers have brought up different concerns, the author's have provided detailed responses with additional experimental results. I recommend accepting this paper for two main reasons: 1) the authors seem to have addressed the majority of concerns in their response, 2) the fact that "online on-policy algorithms are naturally able to solve offline RL problems" is a finding and insight that is likely of interest to the community, even if reviewers may feel that the method is not strongly superior in performance to more standard offline RL algorithms. For the latter, this is even likelier, given that the paper appears to be very well-written.

**Note From Pc:**

if the above contains the word "oral" or "spotlight" please see: "oral" presentation means -> notable-top-5% and "spotlight" means -> notable-top-25%. As stated in our emails, we are disassociating presentation type from AC recommendations